



# Interplay of temperature, productivity, and community assemblage on hydrogen isotope signatures of algal lipid biomarkers

S. Nemiah Ladd[1], Nathalie Dubois[1,2], Carsten J. Schubert[1,3]

[1]Department of Surface Waters – Research and Management, Eawag, Swiss Federal Institute of Aquatic Science and
Technology, Kastanienbaum, 6047, Switzerland
[2]Department of Earth Sciences, ETH Zürich, Zürich, 8092, Switzerland
[3]Institute of Biogeochemistry and Pollutant Dynamics, ETH Zürich, Zürich, 8092, Switzerland

*Correspondence to*: S. Nemiah Ladd (Nemiah.Ladd@eawag.ch)

**Abstract.** The hydrogen isotope composition ($\delta^2$H) of biomarkers produced by algae is strongly influenced by the
$\delta^2$H values of the water in which they grew. $\delta^2$H values of algal biomarkers preserved in lake sediments are thus a useful tool for reconstructing past changes in lake water $\delta^2$H values, which can be used to infer hydroclimate variability. However, studies from laboratory cultures of marine algae have shown that a number of factors can influence the magnitude of hydrogen isotope fractionation between algal lipids and their source water, including temperature and growth rates. Quantifying the natural extent of these changes in freshwater lacustrine settings and identifying their causes is essential for
robust application of $\delta^2$H values of algal lipids as paleohydroclimate proxies, yet the influence of these factors remains poorly constrained.

This work targets the effect of temperature and productivity on $^2$H/$^1$H fractionation in algal biomarkers through a comparative time series in two central Swiss lakes: eutrophic Lake Greifen and oligotrophic Lake Lucerne. Particulate organic matter was collected from surface waters at six time points throughout the spring and summer of 2015, and $\delta^2$H
values of short chain fatty acids, as well as the diatom biomarker brassicasterol, were measured. We paired these measurements with in situ incubations conducted with NaH$^{13}$CO$_3$, which were used to calculate the production rates of individual lipids in lake surface water. As algal productivity increased from April to June, the magnitude of $^2$H/$^1$H fractionation in Lake Greifen increased by as much as 148 ‰ for individual fatty acids. During the same time period in Lake Lucerne, the magnitude of $^2$H/$^1$H fractionation increased by as much as 58 ‰ for individual fatty acids, consistent with the 2
– 4 ‰ per °C increase in $^2$H/$^1$H fractionation observed in cultures of microalgae. Larger changes in $^2$H/$^1$H fractionation in Lake Greifen may be due to a combined effect of higher temperatures and increased algal productivity, which can cause relatively greater contributions of highly depleted H from NADPH in Photosystem I to be incorporated into lipids, or due to seasonal changes in the structure of the algal community. Fatty acid $\delta^2$H values were preserved in surface sediment, while those of brassicasterol indicated large (~40 ‰) isotopic effects due to degradation. The magnitude of brassicasterol
fractionation was significantly different between the two lakes, suggesting that its hydrogen isotope composition may be more sensitive to nutrient regime than that of fatty acids.



# 1 Introduction

The stable isotopic composition of lake water is determined by climatic factors including temperature, moisture source, and the balance of precipitation to evaporation (Craig and Gordon, 1965; Gat, 1996; Henderson and Schuman, 2009; Steinmann et al., 2013). As such, reconstructions of water isotopes are useful for understanding past changes in hydroclimate. Since the

water itself is no longer available, isotopic measurements of materials preserved in lake sediments must be analyzed to deduce changes in water isotopes. Oxygen isotopes of authigenic carbonates have been particularly helpful in this regard (Talbot, 1990; McKenzie and Hollander, 1993; Leng and Marshall, 2004; Nelson et al., 2011; Bird et al., 2011).

However, carbonates are not preserved in all sedimentary archives, and they can precipitate under disequilibrium with

surface water (Fronval et al., 1995; Teranes et al., 1999; Leng and Marshall, 2004). The hydrogen isotopic composition of lipid biomarkers produced by photoautotrophs growing within the lake represents another promising tool for reconstructing surface water isotopes (Sessions et al., 1999; Sauer et al., 2001; Huang et al., 2004; Sachse et al. 2012; Sachs, 2014). The hydrogen isotopic composition of algal lipids is well correlated with those of source water in laboratory and field settings (Sauer et al., 2001; Huang et al., 2004; Englebrecht and Sachs, 2005; Zhang and Sachs, 2007; Sachse et al., 2012), and is

stable under near surface temperatures and pressures for carbon-bound hydrogen (Sessions et al., 2004; Schimmelmann et al., 2006). Algal biomarker hydrogen isotopes have been successfully applied to infer changes in past climate using sediment cores from diverse lakes (Huang et al., 2002; Sachs et al., 2009; Smittenberg et al., 2011; Atwood and Sachs, 2014; Zhang et al., 2014; Nelson and Sachs, 2016; Richey and Sachs, 2016; Randlett et al., 2017) and marine settings (Pahnke et al., 2007; van der Meer et al., 2007; van der Meer et al., 2008; Leduc et al, 2013; Vasiliev et al., 2013; Kasper et al., 2014; Vasiliev et

al., 2017).

The hydrogen isotopic composition of lipids (expressed as $\delta^2H_{Lipid}$, where $\delta^2H = (^2H/^1H_{Sample})/(^2H/^1H_{VSMOW}) + 1$) is not equal to the hydrogen isotopic composition of water, $\delta^2H_{Water}$. The offset between $\delta^2H_{Lipid}$ and $\delta^2H_{Water}$ is described by the apparent fractionation factor, $\alpha_{Lipid\text{-}Water} = (^2H/^1H_{Lipid})/(^2H/^1H_{Water})$. There is increasing evidence that $\alpha_{Lipid\text{-}Water}$ is not constant, and can

change with variables such as salinity, algal species, light availability, growth rate, and temperature (Summarized in Table 1) (Sachs, 2014 and sources therein; Chivall et al., 2014; M'boule et al., 2014; Nelson and Sachs, 2014; Heinzelmann et al., 2015; Sachs and Kawka, 2015; van der Meer et al., 2015; Wolhowe et al., 2015; Maloney et al., 2016; Sachs et al., 2016). While the array of secondary isotope effects can appear daunting, these relationships can provide useful information about past environmental changes in their own right, and developing a thorough understanding of them is important for robust

interpretations of sedimentary $\delta^2H_{Lipid}$ values.

Most previous investigations into variability in $\alpha_{Lipid\text{-}Water}$ in algal lipid biosynthesis have been done with controlled axenic cultures in laboratory settings. For example, temperature has been demonstrated to increase hydrogen isotope fractionation



by 2-4 ‰ in batch cultures of the freshwater green algae *Eudorina unicocca* and *Volvox aureus* (Zhang et al., 2009a), as well as in the marine coccolithophorids *Emiliana huxleyi* and *Gephyrocapsa oceanica* (Wolhowe et al., 2009). In nutrient limited chemostats, hydrogen isotope fractionation increased as growth rate increased for both the marine diatom *Thalassiosira pseudonana* (Zhang et al., 2009a) and for *E. huxleyi* (Kawka and Sachs, 2015). Higher growth rates caused by different

temperature and salinity treatments also resulted in more hydrogen isotope fractionation in batch cultures of *E. huxleyi* and *G. oceanica* (Schouten et al., 2006). Likewise, higher growth rates induced by increased nutrient availability also corresponded to increased hydrogen isotope fractionation in batch cultures of multiple strains of *E. huxleyi* (Wolhowe et al., 2015).

Similar relationships between salinity and hydrogen isotope fractionation have been observed for algal lipids in both laboratory (Schouten et al., 2006; Chivall et al., 2014; M'boule et al., 2014; Heinzelmann et al., 2015; Maloney et al., 2016; Sachs et al., 2016) and field calibrations (Sachse and Sachs, 2008; Sachs and Schwab, 2011; Nelson and Sachs, 2014). The temperature and growth rate effects have yet to be assessed in the lacustrine settings where algal $\delta^2H_{Lipid}$ values are likely to be applied to reconstruct past hydroclimate. In contrast to cultures, lake water contains a diverse and dynamic community of

phytoplankton, most of whom contribute lipids to the sediment that cannot be attributed to one particular species. The culturing data that exist are limited to a few species, many of which are only found in marine environments.

In order to evaluate the significance of temperature and growth rate effects on the hydrogen isotopic composition of algal lipids produced in lakes, we collected monthly samples of particulate organic matter in two lakes in central Switzerland

throughout the spring and summer of 2015. Both lakes experience similar changes in surface water temperature during this time period, but one of them (Lake Greifen) is characterized by high nutrient availability and increasing algal productivity and biomass throughout the spring and early summer. The other lake (Lake Lucerne) is oligotrophic and had relatively low, constant rates of algal productivity throughout the study period. We paired measurements of hydrogen isotope fractionation with in situ incubations designed to determine lipid production rates, allowing us to distinguish between the effects of

productivity and temperature on hydrogen isotope fractionation in algal lipids.

## 2 Methods

### 2.1 Site description

Lake Greifen (Greifensee) is a small perialpine lake, located in the eastern fringes of the Zurich metropolitan area at 47° 21' N and 8° 40' E (Fig. 1). The lake has a surface area of 24 km$^2$, and a maximum depth of 32 m. It is fed by three small

brooks, and has one main outlet, the Glatt Canal. Lake Greifen experienced severe eutrophication in the mid-20$^{th}$ century (Hollander et al., 1992; Keller et al., 2008). Since strict government regulations on nutrient inputs were imposed in the 1970s the water quality in the lake has improved, but its deep water remains anoxic and nutrient levels in the upper water column are still elevated. Winter overturn in the lake brings additional nutrients to the surface water, resulting in large phytoplankton



blooms in the spring and summer as temperature and light availability increase (McKenzie, 1982). All samples from Lake Greifen were collected from the northern part of the lake, near a permanent monitoring platform maintained by Eawag (at 47° 21.99' N, 8° 39.89' E).

Lake Lucerne (Vierwaldstättersee) is a large perialpine lake, located in central Switzerland at 47° 0' N and 8° 30' E (Fig. 1). The lake, which has a total surface area of 116 $km^2$, is formed of seven distinct basins, of which the deepest is 214 m. The lake is fed by four alpine rivers: the Reuss, Muota, Engelberger Aa, and Sarner Aa, and its primary outflow is Reuss river from the northwest tip of the lake. Although Lake Lucerne experienced a mild eutrophication event in the 1970s, it is oligotrophic today (Bürgi et al., 1999; Bührer and Ambühl, 2001; Thevenon et al., 2012). All samples from Lake Lucerne
were collected from the center of Kreuztricher basin (near 8° 21' N, 47° 0' E).

## 2.2 Sample collection

Particulate material in each lake was collected at approximately monthly intervals throughout the spring and summer of 2015 (mid-April through early September). Surface water (~0.5 m water depth) was filtered onto a pre-combusted 142 mm
diameter GF/F filter (0.7 μm pore size) using a WTS-LV Large Volume Pump (McLane, Massachusetts, USA). Pumping began at 7 L/min and continued until the flow rate decreased to 4 L/min or until 25 minutes had passed. All filters were collected at midday on sunny or mostly sunny days. Filters were wrapped in combusted aluminum foil and stored in a cool box on ice until transport to the laboratory, where they were stored at -20 °C until analysis.

Water samples were collected from surface water before and after pumping began. Samples were collected in 4 mL screw cap vials, sealed with electrical tape, and stored at room temperature prior to analysis. Depth profiles of temperature, conductivity, pH, turbidity, and dissolved oxygen were collected for the upper 20m of the water column each sampling day at the beginning and end of filtration using a multiparameter CTD probe (75M, Sea & Sun Marine Tech, Trappenkamp, Germany).

On the morning of each sampling day, 4 x 12.5 L of surface water was collected in acid-rinsed, autoclaved carboys for in situ incubations. In two of the four carboys, 1 mL of concentrated $NaH^{13}CO_3$ solution was added. The other two carboys did not receive this isotopic label. Carboys were mixed and attached to a fixed, floating line so that they stayed in the upper 50 cm of lake water throughout the day. After 6 hours, they were retrieved and the contents were filtered onto a pre-combusted 142
mm diameter GF/F filter using a peristaltic pump. Water samples for DIC analyses were collected in 12 mL exetainers prior to isotopic labeling, after labeling but before incubation, and after incubation. These samples were sterile filtered through a 0.2 μm syringe filter and stored in the dark at 4 °C prior to analysis.



Sediment traps consisting of two open tubes with an active surface area of 130 cm$^2$ and a height of 76 cm (EAWAG-130) were deployed at a water depth of 20m from mid-April to mid-May in both lakes. Material that accumulated in the traps was poured into a jar and kept frozen at -20 °C prior to analysis. Surface sediment was collected in the spring of 2015 with a gravity corer. The upper 1 cm of sediment was sectioned and stored frozen at -20 °C prior to analysis. Material from the

upper 30 cm of each core was analyzed using a Ge Gamma Spectrometer (GCW3022-7500, Canberra, Meridian, CT, USA) on the 46.5 keV line for $^{210}$Pb and the 661.7 keV line for $^{137}$Cs activity in order to determine sediment accumulation rates in each lake, which were ~0.25 cm/year in Lake Lucerne and ~0.3 cm/year in Lake Greifen (S1).

## 2.3 Water isotope measurements

Surface water isotope samples were filtered through a 25 mm syringe filter with a 0.45 μm polyethersulfone membrane to remove particulate matter. Water $\delta^2$H and $\delta^{18}$O values were measured by Cavity Ring Down Spectroscopy (CRDS) on a L-2120i Water Isotope Analyzer (Picarro, Santa Clara, CA, USA) at ETH-Zurich. Each sample was injected seven times in sequence, and the first four values were discarded to avoid any memory effects from the previous sample. Three water standards with known $\delta^2$H values of ranging from -161 ‰ to 7 ‰ and $\delta^{18}$O values ranging from -22.5 ‰ to 0.9 ‰ were

injected at the beginning and end of each sequence, as well as after every 10 samples. These standards were used to correct measured values to the VSMOW scale and to account for any instrumental drift over the course of the sequence. Average standard deviations were 0.4 ‰ for hydrogen isotopes and 0.06 ‰ for oxygen isotopes.

## 2.4 DIC concentrations and $\delta^{13}$C measurements

DIC concentrations were measured on a TOC-L$_{CSH/CHN}$ Total Organic Carbon Analyzer (Shimadzu, Kyoto, Japan). Solutions with DIC concentrations ranging from 5 mg/L to 100 mg/L were injected at the beginning of the sequence to form a calibration curve, and one standard of 50 mg/L was run after every five samples. Samples were analyzed in triplicate.

3.7 mL exetainers were prepared for $\delta^{13}$C measurements of DIC by adding 100 μL of concentrated H$_3$PO$_4$ and filling the

headspace with He. 1 mL of sample water added with a syringe through the septa of the exetainer. Samples were allowed to equilibrate overnight before analysis. Carbon isotope values were measured on an Isotope Ratio Mass Spectrometer (IRMS) (Isoprime, Stockport, United Kingdom). A standard of known isotopic composition was analyzed after every 6 samples. All samples were measured in duplicate.

## 30  2.5 Lipid extraction and purification

An internal standard containing $n$C$_{19}$-alkanol, $n$C$_{19}$-alkanoic acid, and 5α-cholestane was quantitatively added to freeze-dried filters, which were extracted in 30 mL of 9:1 Dichloromethane (DCM)/Methanol (MeOH) in a SOLVpro Microwave Reaction System (Anton Paar, Graz, Austria) at 70 °C for 5 minutes, centrifuged, and the supernatant containing the total lipid extract (TLE) was poured off and evaporated under a gentle stream of N$_2$. The TLE was saponified with 3 mL of 1 N




KOH in MeOH and 2 mL of solvent-extracted nanopure $H_2O$ for 3 hours at 80 °C, after which the neutral fraction was extracted with hexane. Subsequently, the aqueous phase was acidified to pH = 2, and the protonated fatty acids were extracted with hexane.

Neutral fractions were further purified using silica gel column chromatography. The sample was dissolved in hexane and loaded onto a 500 mg/6 mL Isolute Si gel column (Biotage, Uppsala, Sweden). *N*-alkanes were eluted in 4 mL of hexane, aldehydes and ketones in 4 mL of 1:1 Hexane/DCM, alcohols in 4 mL of 19:1 DCM/MeOH, and remaining polar compounds in 4 mL of MeOH. The alcohol fraction was acetylated with 25 μL of acetic anhydride and 25 μL of pyridine for 30 minutes at 70 °C. The $\delta^2H$ and $\delta^{13}C$ values of the added acetyl group were determined by analyzing acetylated and
unacetylated $n$C$_{12}$-alkanol.

Further purification was necessary in order to obtain base line separation of brassicasterol (24-methyl cholest-5,22-dien-3β-ol) for $\delta^2H$ measurements. This was achieved by loading the acetylated alcohol fraction onto 500 mg of Si gel impregnated with AgNO$_3$ (10% by weight, Sigma Aldrich) in a 6 mL glass cartridge. The first fraction was eluted with 20 mL of 4:1
hexane/DCM, the second fraction with 20 mL of 1:1 hexane/DCM, the third fraction (containing brassicasterol) with 16 mL of DCM, and the fourth fraction with 4 mL of ethyl acetate.

Fatty acid fractions were methylated with 1 mL of BF$_3$ in MeOH (14% by volume, Sigma Aldrich) for 2 hours at 100 °C. After methylation, 2 mL of nanopure $H_2O$ was added to the sample and the fatty acid methyl esters (FAMEs) were extracted
with hexane. The $\delta^2H$ and $\delta^{13}C$ values of the added methyl group were determined by methylating phthalic acid of known isotopic composition (Arndt Schimmelmann, Indiana University).

FAMEs and brassicaterol were quantified by gas chromatography – flame ionization detection (GC-FID) (Shimazdu, Kyoto, Japan). Samples were injected by an AOC-20i autosampler (Shimadzu) through a split/splitless injector operated in splitless
mode at 280 °C. The GC column was an InertCap 5MS/NP (0.25 mm x 30 m x 0.25 μm) (GL Sciences, Japan) and it was heated from 70 °C to 130 °C at 20 °C/min, then to 320 °C at 4 °C/min, and held at 320 °C for 20 minutes. FAMEs were identified by comparing their retention times to an external standard (Fatty Acid Methyl Ester mix from Sulpelco, Ref 47885-U). Brassicasterol was identified by comparing its retention time to that obtained by analyzing a subset of samples by gas chromatography – mass spectrometry (GC-MS) under identical conditions. In order to determine how much of the
compound was in the original sample, peak areas were normalized to those of the internal standard. Peak areas were quantified relative to an external calibration curve in order to determine suitable injection volumes for isotopic analysis.



## 2.6 Lipid δ²H and δ¹³C measurements

The stable isotope values of individual FAMEs and brassicasterol were measured by gas chromatography – isotope ratio mass spectrometry (GC-IRMS). A GC-1310 gas chromatograph (Thermo Scientific, Bremen, Germany) equipped with an InertCap 5MS/NP (0.25 mm x 30 m x 0.25 μm) (GL Sciences, Japan) was interfaced to a Delta Advantage IRMS (Thermo Scientific) with a Conflow IV interface (Thermo Scientific). Samples were injected with a TriPlusRSH autosampler to a PTV inlet operated in splitless mode at 280 °C. The oven was heated from 80 °C to 215 °C at 15 °C/min, then to 320 °C at 5 °C/min, and then was held at 320 °C for 10 minutes. Hydrogen isotope samples were pyrolyzed at 1420 °C after they eluted from the GC column. Carbon isotope samples were combusted at 1020 °C after elution.

Raw isotope values were initially converted to the VSMOW (hydrogen) and VPDB (carbon) scales using Thermo Isodat 3.0 software and pulses of a reference gas that was measured at the beginning and end of each analysis. Sample δ²H and δ¹³C values were further corrected using the slope and intercept of measured and known values of isotopic standards ($n$C$_{17, 19, 21, 23, 25, 28,}$ and $_{34}$-alkanes, Arndt Schimmelmann, Indiana University), which were run at the beginning and end of each sequence, as well as after every 6 to 8 sample injections. Offsets between measured and known values for these standards were used to correct for any drift over the course of the sequence or any isotope effects associated with peak area or retention time. The standard deviation for these standards averaged 4 ‰ and the average offset from their known values was 2 ‰ for hydrogen isotopes. For carbon isotopes, the average standard deviation of isotopic standards was 0.4 ‰ and the average offset from known values was 0.1 ‰ over the period of analysis.

An additional standard of $n$C$_{29}$-alkane was measured three times per sequence, corrected in the same way as the samples, and used for quality control. The standard deviation of these measurements was 4 ‰ for hydrogen and 0.5 ‰ for carbon over the period of analysis. The H$_3^+$ factor was measured at the beginning of each sequence and averaged 3.6 ± 0.3 during the analysis period. Samples were corrected for hydrogen and carbon added during derivatization using isotopic mass balance.

## 2.7 Calculated lipid production rates

Lipid production rates were calculated using Eq. (1) (modified from Popp et al. 2006):

$$Production\ rate = (\delta^{13}C_l - \delta^{13}C_n)/(\delta^{13}C_{DIC} - \delta^{13}C_n) * (C_t/t) \qquad (1)$$

where $\delta^{13}C_l$ is the δ¹³C value of the target compounds from labeled incubations, $\delta^{13}C_n$ is that from unlabeled incubations, $\delta^{13}C_{DIC}$ is the δ¹³C value of DIC, $C_t$ is the concentration of the lipid at the end of the incubation, and $t$ is the duration of the incubation. Residence times – assuming a steady state, the amount of time needed to replace all molecules of a given lipids – were calculated by dividing $C_t$ by the production rate, which reduces to Eq. (2):



$$Residence\ time = t*(\delta^{13}C_{DIC} - \delta^{13}C_n)/(\delta^{13}C_l - \delta^{13}C_n) \qquad (2)$$

**2.8 Statistics**

PRISM software (Graphpad Software Inc., La Jolla, CA, USA) was used to carry out all statistical analyses. Ordinary least
squares regression was used to determine relationships between fractionation factors and temperature, fractionation factors
and lipid production rates. Regression lines are only shown in figures where the slope of the regression was significantly
different from 0 at the $p < 0.05$ level. The results of all linear regression analyses are presented in Table 2. Differences
between the slopes of various regressions were assessed using a two-tailed test of the null hypothesis that both slopes are
equal. Differences in the mean values of replicate measurements were determined using an unpaired, two-tailed t-test, and
were considered significantly different for $p < 0.05$.

**3. Results**

**3.1 Lipid concentrations and production rates**

Lipid concentrations increased from low levels in Lake Greifen in April to July, after which they declined slightly (Fig. 2a).
$nC_{16:0}$ fatty acid had the highest concentrations, ranging from $1.0 \pm 0.2$ µg/L in April to $55 \pm 4$ µg/L in July. Concentrations
for $nC_{16:1}$ fatty acid were usually lower than those of other fatty acids, ranging from $1.1 \pm 0.3$ µg/L in April to $13 \pm 2$ µg/L in
September. Brassicaterol (24-methyl cholest-5,22-dien-3β-ol) concentrations were 1-2 orders of magnitude smaller than
those of fatty acids, ranging from $0.18 \pm 0.06$ µg/L in April to $0.9 \pm 0.2$ µg/L in July (Fig. 2a).

Lipid concentrations in Lake Lucerne were generally lower and more stable throughout the summer than in Lake Greifen
(Fig. 2b). Again, $nC_{16:0}$ fatty acid had the highest concentrations, ranging from $0.80 \pm 0.03$ µg/L in April to $15 \pm 3$ µg/L in
early August. $nC_{16:1}$ was the least abundant fatty acid, ranging from $0.80 \pm 0.02$ µg/L in April to $5 \pm 2$ µg/L in early August.
Brassicasterol concentrations were also significantly lower than those of fatty acids in Lake Lucerne, ranging from $0.05 \pm$
$0.01$ µg/L in April to $0.15 \pm 0.03$ µg/L in June (Fig. 2b).

In both lakes, fatty acid production rates were highest for $nC_{16:0}$, followed by $nC_{18:x}$ (unsaturated $C_{18}$ fatty acids, primarily
$nC_{18:1n9c}$, or oleic acid), $nC_{14:0}$, and $nC_{16:1}$ (palmitoleic acid) (Fig. 2c and 2d). Brassicasterol production rates were ~ 3 orders
of magnitude lower in both lakes than those of fatty acids (Fig. 2c and 2d). Lipid production rates were up to three times
higher in Lake Greifen than in Lake Lucerne (Fig. 2c and 2d). Lipid production rates generally increased from May to July
and then remained high in Lake Greifen, while in Lake Lucerne they were relatively constant throughout the study period
(Fig. 2c and 2d).



Residence times – or the amount of time necessary to replace all molecules of a given compound assuming steady state – of individual lipids were calculated according to Eq. (2) (Sect. 2.7) and were typically lowest for $nC_{14:0}$, $nC_{16:0}$, and $nC_{18:x}$ fatty acids, with values as low as $9 \pm 1$ h in Lake Greifen in May, and as low as $12 \pm 3$ h in Lake Lucerne in August (Table 3). Of the fatty acids, $nC_{16:1}$ had the longest residence times, reaching $60 \pm 13$ h in Lake Greifen in May and $62 \pm 27$ h in Lake

Lucerne in June (Table 3). Brassicasterol residence times were the longest of any lipid, and were as long as $258 \pm 61$ h in Lake Lucerne in August and $165 \pm 51$ h in Lake Greifen in July (Table 3).

### 3.2 Lipid $\delta^2$H and $\alpha_{\text{lipid-water}}$ values

In both lakes lipid $\delta^2$H values typically decreased over the spring and summer (Fig. 2e and 2f). This effect was more

pronounced in Lake Greifen than in Lake Lucerne. For example, $nC_{16:0}$ fatty acid $\delta^2$H values declined by 133 ‰ (from -172 ‰ to -305 ‰) from April to August in Lake Greifen, while they only declined by 53 ‰ (from -249 ‰ to -302 ‰) over the same time period in Lake Lucerne (Fig. 2e and 2f). During the same time period, water $\delta^2$H values increased slightly in Lake Greifen (from -73 ‰ to -65 ‰) and were relatively constant in Lake Lucerne (fluctuating between -82 ‰ and -86 ‰) (S2). Changes in the magnitude of hydrogen isotope fractionation between fatty acids and surface water ($\alpha_{\text{lipid-water}}$) were therefore

primarily due to changes in fatty acid $\delta^2$H values. In Lake Greifen, $\alpha_{\text{lipid-water}}$ for $nC_{16:0}$-fatty acid decreased from 0.891 to 0.743 from April to August (Fig. 2g), while in Lake Lucerne, it decreased from 0.821 to 0.763 (Fig. 2h). Similar patterns were observed for $\alpha_{\text{lipid-water}}$ for $nC_{14:0}$, $nC_{16:1}$ and $nC_{18:X}$ fatty acids (Fig. 2g and 2h). Values for $\alpha_{\text{lipid-water}}$ were less variable for brassicasterol than for fatty acids. Brassicasterol $\delta^2$H values were always depleted relative to those of fatty acids in Lake Greifen, and were depleted to all fatty acids except $nC_{14:0}$ in Lake Lucerne (Fig. 2g and 2h).

Overall, fatty acid $\alpha_{\text{lipid-water}}$ values were negatively correlated with lake surface temperature in both lakes ($R^2 = 0.32$, p = 0.004 in Lake Greifen; $R^2 = 0.24$, p = 0.01 in Lake Lucerne) (Fig. 3a; Table 2). The slope of the relationship was significantly steeper (p = 0.03) in Lake Greifen than in Lake Lucerne (m = -0.006 $\pm$ 0.002 in Lake Greifen and -0.003 $\pm$ 0.001 in Lake Lucerne). When analyzing individual fatty acids, rather than all short-chain fatty acids, $R^2$ values for the

negative correlation were typically higher (Fig. 3; Table 2). Significant correlations were observed between lake surface temperature and $\alpha_{\text{lipid-water}}$ values for most fatty acids, but not for $nC_{16:1}$ in Lake Greifen and $nC_{14:0}$ in Lake Lucerne (Fig. 3; Table 2). Significant relationships between lake surface temperature and $\alpha_{\text{lipid-water}}$ values were not observed in either lake for brassicasterol (Fig. 3f; Table 2).

Fatty acid production rates were not correlated with $\alpha_{\text{lipid-water}}$ values in either lake (Table 2). Among individual fatty acids, only $nC_{16:1}$ fatty acids from Lake Greifen had a significant negative correlation between $\alpha_{\text{lipid-water}}$ values and production rate ($R^2 = 0.84$; p = 0.03) (Table 2). Brassicasterol $\alpha_{\text{lipid-water}}$ values were not correlated with production rates in either lake (Table 2), although values from less productive Lake Lucerne cluster as a significantly higher group than in Lake Greifen (p = 0.0004).




### 3.3 Sedimentary lipid δ²H values

Lipid $\delta^2H$ values in surface sediment and sediment traps from each lake displayed less variability than those in surface water particulate matter (Fig. 4). For $n$C$_{16:0}$-fatty acid, sedimentary $\delta^2H$ values (-249 ± 3 ‰ in Lake Greifen; -280 ± 1 ‰ in Lake

Lucerne) were not significantly different from mean surface water particulate matter $\delta^2H$ values in either lake (-250 ± 22‰ in Lake Greifen; -279 ± 8 ‰ in Lake Lucerne) (Fig. 4a). For brassicasterol, sedimentary $\delta^2H$ values were enriched relative to surface water values by 36 ‰ in Lake Greifen and by 40 ‰ in Lake Lucerne (Fig. 4b).

## 4. Discussion

### 4.1 Factors influencing hydrogen isotope fractionation in short-chain fatty acids

Several factors have been shown to influence hydrogen isotope fractionation in algae in laboratory settings, including salinity, light availability, temperature, growth rate, and species (Summarized in Table 1) (Sachs, 2014 and sources therein; Chivall et al., 2014; M'boule et al., 2014; Nelson and Sachs, 2014; Heinzelmann et al., 2015; Sachs and Kawka, 2015; van

der Meer et al., 2015; Wolhowe et al., 2015; Maloney et al., 2016; Sachs et al., 2016). Of these, salinity can be excluded as a source of variability in the freshwater Lakes Greifen and Lucerne. The likely influence of temperature, light availability, productivity, and composition of the algal community on the seasonal changes in hydrogen isotope fractionation for short-chain fatty acids in Lakes Greifen and Lucerne are explored below.

**4.1.1 Temperature**

The results from laboratory cultures demonstrate that higher temperatures can cause hydrogen isotope fractionation in algae to increase by 2 - 4 ‰ per °C (Schouten et al., 2006; Zhang et al., 2009a; Wolhowe et al., 2009; Wolhowe et al., 2015). Increased hydrogen isotope fractionation at higher temperatures has been attributed to (*i*) changes in the relative activity of different enzymes involved in lipid synthesis at different temperatures, (*ii*) changes in the relative amount of NADPH from

the pentose phosphate cycle as temperature changes, and (*iii*) the potential for hydrogen tunneling at higher temperatures as substrate-enzyme complex vibrations increase (Sachs, 2014 and references therein).

The relationship between temperature and hydrogen isotope fractionation in cultures is similar to that observed in Lake Lucerne, where $\alpha_{\text{lipid-water}}$ decreases by 0.003 ± 0.001 per °C (equivalent to a 3 ± 1 ‰ increase in fractionation per °C) (Fig. 3;

Table 2). The relationship between $\alpha_{\text{lipid-water}}$ and temperature in Lake Greifen (-6 ± 2 ‰ per °C) (Fig. 3; Table 2) is much steeper than that observed in culturing studies. The slope of the relationship between $\alpha_{\text{lipid-water}}$ and temperature is similar when plotting all fatty acids for each lake together or separated by chain length (Fig. 3; Table 2), although the correlations are generally better when plotting individual lipids separately, as they are each associated with a different net fractionation factors relative to each other (Fig. 2 g-h).



If the influence of temperature on algal hydrogen isotope fractionation is consistent among laboratory cultures and lakes, warmer temperatures can account for the entire seasonal change in $\alpha_{\text{lipid-water}}$ for fatty acids in Lake Lucerne. However, increasing temperatures can explain only part of the decrease in fatty acid $\delta^2H$ values in Lake Greifen over the course of the spring and summer. Temperature could account for as much as half of the change in $\alpha_{\text{lipid-water}}$ in Lake Greifen, assuming a consistent relationship to that observed in cultures. An additional factor (or factors), such as greater changes in productivity over the course of the summer, or different changes in the composition of the algal community, is necessary to explain the large shift in hydrogen isotope fractionation over the course of the summer in Lake Greifen.

### 4.1.2 Light availability

The influence of light availability on $\alpha_{\text{lipid-water}}$ has never been investigated for freshwater algae, although some evidence exists to suggest that low light levels can increase hydrogen isotope fractionation in marine coccolithophorids. Hydrogen isotope fractionation of alkenones decreased by ~40 ‰ when light levels were increased from 15 to 200 µmol photons $m^{-2}$ $s^{-1}$ in cultures of *E. huxleyi* (van der Meer et al., 2015). Likewise, hydrogen isotope fractionation was observed to decrease by ~50 ‰ for alkenones produced at different depths in the Gulf of California and the Eastern Tropical North Pacific as noon photosynthetically available light (PAR) increased from 10 to 250 µmol photons $m^{-2}$ $s^{-1}$ (Wolhowe et al., 2015). In both of these studies, further increases in light intensity did not result in any change in $\alpha_{\text{lipid-water}}$.

PAR was not measured as part of the present study. However, as all of our samples were collected from lake surface water at a mid-latitude northern hemisphere site during boreal spring and summer, it is unlikely that PAR was less than 250 µmol photons $m^{-2}$ $s^{-1}$ at any sampling date (Pinker and Laszlo, 1992), meaning that the effect of light intensity is unlikely to be a source of the observed seasonal variability in $\alpha_{\text{lipid-water}}$ in lake surface water. To date, no studies have investigated the possibility for hours of daylight to influence hydrogen isotope fractionation in algae, and daylight hours clearly changed throughout the study period. Given that greater light intensity leads to less hydrogen isotope fractionation for alkenones, it might be expected that longer daylight hours would cause less hydrogen isotope fractionation in other algal lipids. The opposite trend was observed as daylight hours increased from April to June in Lakes Greifen and Lucerne, suggesting that light effects were not a significant primary cause of seasonal changes in $\alpha_{\text{lipid-water}}$. However, indirect effects of increased light availability, through the promotion of algal growth, may have had an important seasonal effect on $\alpha_{\text{lipid-water}}$.

This effect may have been more pronounced in Lake Greifen than in Lake Lucerne, since higher nutrient concentrations in the former allow for higher productivity in response to increasing light availability. Greater changes in productivity in Lake Greifen are indicated by the large increase in lipid concentrations and production rates in this lake (Fig. 2a, 2c) relative to the more static levels in Lake Lucerne (Fig. 2b, 2d), which mirror the change in algal abundance in the two lakes (Fig. 5). Since the two lakes are close to each other and experience similar weather patterns, changes in light availability and daylight hours



cannot directly account for the greater changes in hydrogen isotope fractionation in Lake Greifen compared to Lake Lucerne (Fig. 2g-h).

### 4.1.3 Lipid production rate

Increased algal growth rates have been shown to result in increased hydrogen isotope fractionation during lipid synthesis in laboratory settings (Schouten et al., 2006; Zhang et al., 2009a; Sachs and Kawka, 2015; Wolhowe et al., 2015). This relationship is most likely caused by increased contributions of hydrogen from relatively enriched NADPH from the oxidative pentose phosphate cycle under low-growth, nutrient stressed conditions, at the expense of relatively depleted hydrogen from photosystem I (Schmidt et al., 2003; Sachs and Kawka, 2015). Increasing growth rates throughout the spring

into early summer could therefore also be responsible for some or all of the increase in hydrogen isotope fractionation that occurs in algae growing in lake surface water. This effect is expected to be more pronounced in Lake Greifen than in Lake Lucerne, as algae in the latter are more limited by nutrients throughout the spring and summer, and consequently have smaller changes in productivity in response to increasing light availability (Sect. 4.1.2; Fig. 2a-d; Fig. 5).

Lipid concentrations and production rates are not a direct proxy for growth rate, as a higher percentage of algal biomass is typically allocated to lipids when algal cells are growing more slowly due to nutrient limitation (Rossler, 1990; Williams and Laurens, 2010). However, higher lipid production rates for the whole community (rather than on a per cell basis) will also occur under conditions of higher growth rates, as adding more cells and biomass at a higher rate will also add more total lipids, and increasing lipid production rates have been shown to have hydrogen isotope effects that are consistent with

increasing growth rates for alkenones in the eastern Pacific Ocean (Wolhowe et al., 2015). Given that both lipid concentrations and production rates are higher in Lake Greifen than in Lake Lucerne, and that higher algal growth rates are to be expected in a more productive, nutrient replete lake, it seems probable that much of the increase in lipid production rates throughout the time series is due to increasing growth rates. Monthly monitoring of algal cell counts (Fig. 5) is also consistent with increased growth rates during the period of increasing lipid production rates in Lake Greifen.

It is therefore possible that the large decline in $\alpha_{lipid-water}$ for fatty acids in Lake Greifen is due to a combined temperature effect of -2 to -4 ‰ per °C (as described in Sect. 4.1.1) and lower $\alpha_{lipid-water}$ values (more fractionation) due to higher growth rates, as indicated by higher lipid production rates. In Lake Lucerne, on the other hand, lipid production rates and probable growth rates are less variable throughout the time series (Fig. 2d), and therefore the only cause of lower $\alpha_{lipid-water}$ values is

increased temperatures.

The weak to non-existent correlations between $\alpha_{lipid-water}$ and lipid production rates (Table 2) stand in opposition to this line of argument. In fact, the only fatty acid that has a negative correlation between $\alpha_{lipid-water}$ and production rate is $nC_{16:1}$ in Lake Greifen (Table 2). This was the only fatty acid from Lake Greifen for which $\alpha_{lipid-water}$ and temperature were not correlated,



which would suggest that an additive effect of temperature and productivity is less likely to explain the steeply negative relationship between $\alpha_{lipid-water}$ and temperature for the other fatty acids in Lake Greifen.

### 4.1.4 Algal community

Another possible source of variability in $\alpha_{lipid-water}$ over the course of the summer could be seasonal changes in the algal community composition. Hydrogen isotope fractionation for $n\text{C}_{16:0}$ fatty acid has been demonstrated to vary by as much as 160 ‰ among five different species of freshwater green algae grown under identical conditions in the laboratory (Zhang and Sachs, 2007). Such variations may be due to different enzymes involved in lipid synthesis among different species. Since there is limited data from culturing experiments, it is not possible to say how widespread such interspecies variability is. It is possible that most algal species display similar magnitudes of hydrogen isotope fractionation during fatty acid synthesis under similar conditions. However, it is equally possible that the fatty acid synthesis varies randomly by hundreds of per mil among species.

Given this uncertainty, and the significant changes in abundance of different algal taxa in Lake Greifen over the course of 2015 (Fig. 5a), contributions of short-chain fatty acids from different species of algae with different magnitudes of hydrogen isotope fractionation could account for some or all of the seasonal variability in $\alpha_{lipid-water}$. A comparable data set of algal species counts does not exist for Lake Lucerne from 2015, but bi-monthly data has been compiled from 2014 (Fig. 5b). Some changes in relative distributions of taxa are similar between the two lakes; for example, both Lake Greifen in 2015 and Lake Lucerne in 2014 experienced a peak in golden algae (*Chrysophyceae*) in June, and elevated abundance of cyanobacteria in late summer and late autumn (Fig. 5). Other trends differ starkly between the two lakes. Green algae (*Chlorophyceae*) are largely absent from Lake Lucerne, while they make up a significant portion of the algal community in Lake Greifen. Notably, the relative abundance of green algae steadily increased from May to September in Lake Greifen (Fig. 5), during which time $\alpha_{lipid-water}$ values declined at a greater rate for most fatty acids than they did in Lake Lucerne (Fig. 2). If green algae tend to have lower $\alpha_{lipid-water}$ values than other algal taxa, their greater abundance in Lake Greifen throughout the summer could account for the greater decline in fatty acid fractionation factors over the course of the time series than in Lake Lucerne.

Another striking difference in the algal community composition between the two lakes is the greater prevalence of golden algae in Lake Lucerne. Many species of golden algae are known to be mixotrophic, meaning that they can employ both photoautrophy and heterotrophy (Flynn et al., 2012), and they tend to be common in low-nutrient environments like Lake Lucerne. Net hydrogen isotope fractionation during the synthesis of fatty acids can differ by as much as 400 ‰ between heterotrophs and photoautophs, with heterotrophs having more enriched fatty acids (Zhang et al., 2009b). If golden algae employing heterotrophy produce a significant portion of the fatty acids in Lake Lucerne's surface waters, one would expect



higher $\alpha_{lipid-water}$ values than in Lake Greifen. This is not the case, with similar values for $\alpha_{lipid-water}$ between the two lakes for most of the summer, and higher values in Lake Greifen in the spring.

An additional possibility is that the samples from mid-April, which are most isotopically distinct from other months (Fig. 2e-f), are influenced by material from heterotrophic bacteria, or mixotrophs relying primarily on heterotrophy. In this scenario, fatty acids from heterotrophs build up in the lake surface during the winter months when photoautotrophy is less viable, and dilute the net fatty acid $\delta^2H$ values with highly enriched material. As phytoplankton productivity ramps up with warmer temperatures, surface water stratification, and longer daylight hours in the spring, newly produced fatty acids from photoautotrophs cause the net fatty acid $\delta^2H$ values to decrease.

For Lake Greifen, a simple isotopic mass balance indicates that 35% of the total $nC_{16:0}$ fatty acid would need to come from heterotrophs with $\alpha_{lipid-water}$ values of 1.150 (the maximum observed by Zhang et al., 2009b) in mid-April if the remaining $nC_{16:0}$ fatty acid was derived from phytoplankton with $\alpha_{lipid-water}$ values of 0.750 (the lowest values observed for photoautotrophs by Zhang et al., 2009b). By the end of the summer, $nC_{16:0}$ fatty acid $\delta^2H$ values are consistent with an exclusive phytoplankton source. Similar calculations suggest that 18% of $nC_{16:0}$ fatty acid would need to come from heterotrophic bacteria in mid-April in Lake Lucerne in order to account for the 50 ‰ decrease in $nC_{16:0}$ fatty acid $\delta^2H$ values over the course of the summer. If the source of so much $nC_{16:0}$ fatty acids was bacterial, it would be expected to correspond to increased concentrations of lipids associated with heterotrophic bacteria, such as iso- and anteiso-$C_{15:0}$ and $C_{17:0}$ (Perry et al., 1979; Volkmann et al., 1980). Since there are not significant amounts of these short-chain odd-carbon branched fatty acids in the particulate organic matter throughout the time series, including in the early spring, it seems unlikely that such a large component of the even-carbon fatty acids could be derived from bacterial sources, although this does not rule out greater contributions from mixotrophic algae relying on heterotrophy in the spring.

## 4.2 Factors influencing hydrogen isotope fractionation in brassicasterol

Brassicasterol (24-methyl cholest-5,22-dien-3β-ol) is a sterol that is commonly used as a biomarker for diatoms, although it has also been detected in some non-diatom algal sources (Volkman et al., 1998; Volkman, 2003; Rampen et al.,2010) and occasionally in plant oils (Zarrouk et al., 2009). Brassicasterol and other sterols closely associated with algal sources have been proposed to be a better target molecule for compound specific hydrogen isotope reconstructions of lake water isotopes, since they are limited to production within the lake, as opposed to short-chain saturated fatty acids, which are also produced in high abundance by trees and other higher plants. Although sterols are generally not restricted to a single species, the number of species responsible for producing them is more limited than for saturated fatty acids, meaning their $\delta^2H$ values might be less susceptible to changes in algal community composition (Sachs, 2014).




Brassicasterol $\alpha_{\text{lipid-water}}$ values do not have a strong relationship with lake surface temperature. There is no correlation between lake surface temperature and $\alpha_{\text{lipid-water}}$ values in Lake Lucerne. In Lake Greifen the two variables are negatively correlated, although the relationship is not quite significant (Fig. 3f; Table 2). The slope of the relationship between $\alpha_{\text{Brassicasterol-water}}$ and lake surface temperature in Lake Greifen is significantly shallower than that observed for fatty acids (-

0.002 ± 0.001 for brassicasterol vs. -0.008 ± 0.002 for $n\text{C}_{16:0}$ fatty acid). Given the lack of relationship between temperature and $\alpha_{\text{Brassicasterol-water}}$ in Lake Lucerne, and the shallow slope in Lake Greifen, it seems unlikely that temperature influences hydrogen isotope fractionation of brassicasterol. If this is the case, the relationship between temperature and $\alpha_{\text{lipid-water}}$ for fatty acids may be best explained by changes in the activity of enzymes involved in the synthesis of fatty acid synthesis and other acetogic lipids, but not in the synthesis of isoprenoidal lipids including sterols.

There is also no correlation between brassicasterol production rates and $\alpha_{\text{Brassicasterol-water}}$ in either lake (Table 2). However, there is a clear, significant difference of 45 ± 8 ‰ (p = 0.0004) in $\alpha_{\text{Brassicasterol-water}}$ between the two lakes (Fig. 4), with lower $\alpha_{\text{Brassicasterol-water}}$ values in Lake Greifen (0.712 ± 0.006) than in the less productive Lake Lucerne (0.757 ± 0.005). This result would be consistent with increased hydrogen isotope fractionation (lower α values) for sterols in more nutrient-limited

systems, as predicted by the culturing results of Zhang et al. (2009a).

Since brassicasterol is not produced by bacterial sources, and is not known to be produced by golden algae, it seems unlikely that the 25 ‰ decrease in its $\delta^2\text{H}$ values from April to May in Lake Greifen could be due to heterotrophic contributions during the winter, as suggested for fatty acids in Sect. 4.1.4. However, there is some evidence to suggest that bacterial

degradation of sterols to stanols preferentially removes sterols with lower $\delta^2\text{H}$ values, enriching the remaining pool of sterols in $^2\text{H}$ (Schwab et al., 2015). Lower brassicasterol production in the winter could therefore result in relatively enriched brassicasterol $\delta^2\text{H}$ values by early spring, if a disproportionate amount of isotopically depleted brassicasterol had been removed from the lake surface water.

**4.3 Implications for paleohydroclimate reconstructions**
Despite the significant seasonal range in fatty acid $\delta^2\text{H}$ values in particulate organic matter in the surface water of both lakes, mean sedimentary $\delta^2\text{H}$ values of fatty acids were not significantly different than those produced in the lake (Fig. 4a). This result suggests that sedimentary fatty acid $\delta^2\text{H}$ values reflect mean water isotopes in the lake surface throughout the entire growth period. It also indicates that significant alteration of fatty acid $\delta^2\text{H}$ values during deposition in temperate lakes with

similar conditions to those found in central Switzerland is unlikely, and is consistent with the results of studies comparing $\delta^2\text{H}$ values of fatty acids in particulate organic matter with those in surface sediments in marine settings (Jones et al., 2008; Li et al., 2009). The consistency between $\delta^2\text{H}$ values of short-chain fatty acids in the water column and their underlying sediments increases the viability of using short-chain fatty acid hydrogen isotopes to reconstruct past hydroclimate. It is also encouraging that despite the large differences in nutrient availability and algal community between the two lakes, $\delta^2\text{H}$ values





of $n$C$_{16:0}$ fatty acid in both suspended particles and in sediments in Lake Greifen are enriched relative to those in Lake Lucerne by ~30 ‰, similar to enrichment of surface water in Lake Greifen relative to Lake Lucerne.

For brassicasterol $\delta^2$H values, a different picture emerges. There is less seasonable variability in the $\delta^2$H values of
brassicasterol in surface water particulate organic matter than there is for $n$C$_{16:0}$ fatty acid (Fig. 2e-f, Fig. 4). However, the sedimentary $\delta^2$H values of brassicasterol fall completely outside the range of what is found in the surface water. In both lakes, sedimentary brassicasterol $\delta^2$H values are enriched by ~40 ‰ relative to mean surface water values. This is consistent with results for dinosterol in the Chesapeake Bay (Sachs and Schwab, 2011) and in lakes from Cameroon (Schwab et al., 2015), where the dinoflagellate biomarker dinosterol was found to have higher $\delta^2$H values in surface sediment than in
suspended particles from the overlying water column. Two possible explanations for the offset are seasonal bias in the material accumulating in sediment, or isotopic effects during the degradation of sterols. Since we have a six-month range of suspended particles and sedimentary brassicasterol $\delta^2$H values fall well outside of these values, it makes the seasonal explanation less likely. It seems more probable that sterol hydrogen isotopes are influenced by degradation, which may complicate their interpretation in sedimentary records.

Additionally, for both particulate organic matter in the lake surface and for sediments, $\delta^2$H values of brassicasterol are depleted in Lake Greifen relative to Lake Lucerne, even though water isotopes are enriched in Lake Greifen compared to those of Lake Lucerne (S2). This indicates that the fractionation factor for brassicasterol in Lake Greifen ($\alpha_{Brassicasterol-Water}$ = 0.751 ± 0.002) is ~50 ‰ lower than in Lake Lucerne ($\alpha_{Brassicasterol-Water}$ = 0.801 ± 0.006), meaning that there is significantly
more net $^2$H fractionation in the former. Increased fractionation during brassicasterol synthesis in the more productive lake could be consistent with the increased hydrogen isotope fractionation observed under nutrient replete and higher growth rate conditions in laboratory cultures (Schouten et al., 2006; Zhang et al., 2009a; Wolhowe et al., 2015), and suggests that the changes in the relative difference between sterol $\delta^2$H values and those of $n$C$_{16:0}$ fatty acid could be explored as an indicator of past algal productivity. However, the effect of degradation on sedimentary sterol $\delta^2$H values would need to be better
constrained before such a proxy could be robustly developed.

## 5. Conclusions

We measured $\delta^2$H values of short-chain fatty acids and the diatom biomarker brassicasterol in two lakes in central Switzerland with different trophic states at six time points throughout the spring and summer of 2015. In order to assess the
relative influence of temperature and productivity on $\alpha_{lipid-water}$, we paired our measurements with in situ incubations with $^{13}$C-enriched DIC that allowed us to calculate lipid production rates.

In oligotrophic Lake Lucerne, lipid concentrations and production rates were relatively low and stable throughout the study period (Fig.2b, 2d). The magnitude of hydrogen isotope fractionation for most fatty acids increased by ~50 ‰ as the season

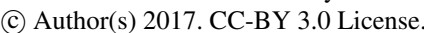


progressed (Fig. 2h). This change is consistent with laboratory cultures that have demonstrated that hydrogen isotope fractionation increases with temperature by 2 – 4 ‰ per °C. Brassicasterol fractionation was not correlated with temperature, suggesting that changes in enzymatic activity are the most plausible of the previously proposed mechanisms to explain increased hydrogen isotope fractionation for acetogenic lipids with higher temperature.

In eutrophic Lake Greifen, the magnitude of hydrogen isotope fractionation for fatty acids increased more dramatically over the study period than in Lake Lucerne (Fig. 2), and the slope of the linear regression between $\alpha_{lipid-water}$ and temperature was two times the maximum value observed in culture (Fig. 3). Changes in $\alpha_{lipid-water}$ in Lake Greifen are best explained as a similar temperature effect to that observed in Lake Lucerne, superimposed on increased fractionation due either to higher
productivity or changes in the algal community structure.

Despite large changes in $\alpha_{lipid-water}$ for $n$C$_{16:0}$ fatty acid in surface particulate matter over the course of the study period, its mean $\delta^2$H values were indistinguishable from those of $n$C$_{16:0}$ fatty acid in surface sediment and sediment traps in both lakes (Fig. 4). The difference in $n$C$_{16:0}$ fatty acid $\delta^2$H values between the lakes was comparable to difference in surface water
isotopes between them, a promising result for the use of fatty acid hydrogen isotopes to reconstruct past hydroclimate. Brassicasterol $\delta^2$H values, on the other hand, were enriched by ~40 ‰ in sediment relative to surface water organic matter, indicating a large hydrogen isotope effect due to degradation. In both sediment and particulate organic matter, brassicasterol $\delta^2$H values indicated hydrogen isotope fractionation that was ~50 ‰ greater in Lake Greifen relative to Lake Lucerne, suggesting that isoprenoid hydrogen isotopes may be more sensitive to nutrient availability than those of fatty acids. If this
signal is preserved despite degradation, it may be possible to develop a proxy for past productivity based on changes in the relative difference between sterol and fatty acid hydrogen isotopes.

**Author contribution**

S. N. Ladd designed the study with input from N. Dubois and C. Schubert. S. N. Ladd and N. Dubois collected the samples.
S. N. Ladd processed and measured the samples. S. N. Ladd, N. Dubois, and C. Schubert contributed to data interpretation. S. N. Ladd prepared the manuscript with contributions from N. Dubois and C. Schubert. The authors declare that they have no conflict of interest.

**Acknowledgements**
This research was funded by a National Science Foundation Earth Sciences Postdoctoral Fellowship (Award #1452254) to NL. Alois Zwyssig and Alfred Lück assisted with sample collection. Serge Robert and Julian Stauffer assisted with sample preparation and laboratory analyses. Daniel Montluçon at ETH-Zurich measured the water isotopes. Algal counts were conducted by Esther Keller as part of Eawag's Department of Aquatic Ecology's long-term monitoring program. We had



productive conversations with Ashley Maloney, Daniel Nelson, Julian Sachs, and Romana Limberger that improved the study design and interpretation of results. We are grateful for all of their contributions.

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





**Table 1** Summary of expected changes in $\alpha_{\text{Lipid-Water}}$ in response to different environmental variables, based on laboratory cultures and field studies in marine settings

| Variable | Sign of correlation with $\alpha_{\text{Lipid-Water}}$ | Magnitude | References |
|---|---|---|---|
| Temperature | Negative | 2 – 4 ‰ per °C | Zhang et al., 2009a; Wolhowe et al., 2009 |
| Growth Rate | Negative | ~30 ‰ per division day$^{-1}$ | Schouten et al., 2006; Zhang et al., 2009a; Sachs and Kawka, 2015; Wolhowe et al., 2015 |
| Nutrient Availability | Negative | ~40 ‰ difference between nutrient limited and nutrient replete cultures | Zhang et al., 2009a; Wolhowe et al., 2015 |
| Light Availability | Positive | Below ~250 μmol photons m$^{-2}$ s$^{-1}$, ~0.2‰ per μmol photons m$^{-2}$ s$^{-1}$ | van der Meer et al., 2015; Wolhowe et al., 2015 |
| Salinity | Positive | 0.5 – 3 ‰ per practical salinity unit (PSU) | Schouten et al. 2006; Sachse and Sachs, 2008; Sachs and Schwab, 2011; Chivall et al., 2014; M'boule et al., 2014; Nelson and Sachs, 2014; Heinzelmann et al., 2015; Maloney et al., 2016; Sachs et al., 2016 |
| Species Assemblage | Variable | Differences up to 160 ‰ observed for $n$C16:0 fatty acid among species growing under identical conditions | Schouten et al., 2006; Zhang & Sachs, 2007 |





**Table 2** Summary of linear regression statistics for all water isotopes and various fractionation factors with salinity; bolded relationships are significant at the p < 0.05 level.

| | LAKE GREIFEN | | | | | LAKE LUCERNE | | | | |
|---|---|---|---|---|---|---|---|---|---|---|
| **α Lipid-Water vs. Temperature** | | | | | | | | | | |
| **Lipid** | **Slope** | **y-intercept** | $R^2$ | **p** | **n** | **Slope** | **y-intercept** | $R^2$ | **p** | **n** |
| All Fatty acids | **-0.006 ± 0.002** | **0.95 ± 0.04** | **0.32** | **0.004** | **24** | **-0.003 ± 0.001** | **0.85 ± 0.02** | **0.24** | **0.015** | **24** |
| $n$C14:0 Fatty acid | **-0.004 ± 0.001** | **0.86 ± 0.03** | **0.75** | **0.03** | **6** | -0.0012 ± 0.0009 | 0.80 ± 0.02 | 0.28 | 0.28 | 6 |
| $n$C16:0 Fatty acid | **-0.008 ± 0.002** | **0.98 ± 0.03** | **0.87** | **0.006** | **6** | **-0.003 ± 0.001** | **0.84 ± 0.02** | **0.66** | **0.049** | **6** |
| $n$C16:1 Fatty acid | -0.005 ± 0.003 | 0.98 ± 0.07 | 0.35 | 0.22 | 6 | **-0.004 ± 0.001** | **0.89 ± 0.02** | **0.74** | **0.028** | **6** |
| $n$C18:x Fatty acid | **-0.006 ± 0.0003** | **0.97 ± 0.007** | **0.99** | **<0.0001** | **6** | **-0.004 ± 0.001** | **0.89 ± 0.02** | **0.70** | **0.037** | **6** |
| Brassicasterol | -0.002 ± 0.001 | 0.75 ± 0.02 | 0.63 | 0.11 | 5 | 0 ± 0.001 | 0.76 ± 0.02 | $3*10^{-5}$ | 0.99 | 6 |

| | LAKE GREIFEN | | | | | LAKE LUCERNE | | | | |
|---|---|---|---|---|---|---|---|---|---|---|
| **α Lipid-Water vs. Lipid Production Rate** | | | | | | | | | | |
| **Lipid** | **Slope** | **y-intercept** | $R^2$ | **p** | **n** | **Slope** | **y-intercept** | $R^2$ | **p** | **n** |
| All Fatty acids | -0.01 ± 0.01 | 0.83 ± 0.02 | 0.10 | 0.18 | 20 | -0.04 ± 0.03 | 0.81 ± 0.01 | 0.09 | 0.19 | 20 |
| $n$C14:0 Fatty acid | 0.00 ± 0.02 | 0.76 ± 0.02 | 0.0002 | 0.98 | 5 | -0.2 ± 0.2 | 0.82 ± 0.05 | 0.25 | 0.39 | 5 |
| $n$C16:0 Fatty acid | 0.01 ± 0.02 | 0.78 ± 0.05 | 0.02 | 0.83 | 5 | -0.01 ± 0.04 | 0.83 ± 0.03 | 0.46 | 0.21 | 5 |
| $n$C16:1 Fatty acid | **-0.17 ± 0.04** | **0.89 ± 0.01** | **0.84** | **0.03** | **5** | 0.6 ± 0.6 | 0.76 ± 0.06 | 0.22 | 0.42 | 5 |
| $n$C18:x Fatty acid | -0.01 ± 0.02 | 0.85 ± 0.04 | 0.08 | 0.64 | 5 | -0.09 ± 0.05 | 0.86 ± 0.02 | 0.57 | 0.14 | 5 |
| Brassicasterol | 2 ± 2 | 0.70 ± 0.01 | 0.34 | 0.42 | 4 | 5 ± 43 | 0.76 ± 0.03 | $4*10^{-3}$ | 0.92 | 5 |





**Table 3**: Mean residence times in hours of lipids in lake surface water, calculated according to equation 2

| Date | $n$C14:0 | $n$C16:0 | $n$C16:1 | $n$C18:x | Brassicasterol |
|---|---|---|---|---|---|
| *Lake Greifen* | | | | | |
| May 11, 2015 | 30 ± 9 | 19 ± 4 | 60 ± 13 | 19 ± 3 | 78 ± 27 |
| June 5, 2015 | 9 ± 2 | 9 ± 1 | 21 ± 2 | 9 ± 1 | 58 ± 34 |
| July 2, 2015 | 19 ± 1 | 20 ± 1 | 49 ± 5 | 20 ± 3 | 165 ± 51 |
| Aug. 11, 2015 | 15 ± 3 | 17 ± 2 | 41 ± 5 | 16 ± 2 | 81 ± 34 |
| Sept. 8, 2015 | 10 ± 3 | 12 ± 2 | 31 ± 8 | 12 ± 2 | 76 ± 37 |
| *Lake Lucerne* | | | | | |
| May 13, 2015 | 16 ± 6 | 16 ± 5 | 27 ± 8 | 17 ± 2 | 124 ± 22 |
| June 3, 2015 | 20 ± 12 | 18 ± 6 | 62 ± 27 | 16 ± 4 | 208 ± 56 |
| July 7, 2015 | 19 ± 8 | 19 ± 5 | 34 ± 18 | 17 ± 5 | 114 ± 39 |
| July 31, 2015 | 22 ± 9 | 19 ± 4 | 38 ± 8 | 18 ± 4 | 164 ± 59 |
| Aug. 31, 2015 | 20 ± 4 | 22 ± 3 | 41 ± 6 | 12 ± 3 | 258 ± 61 |





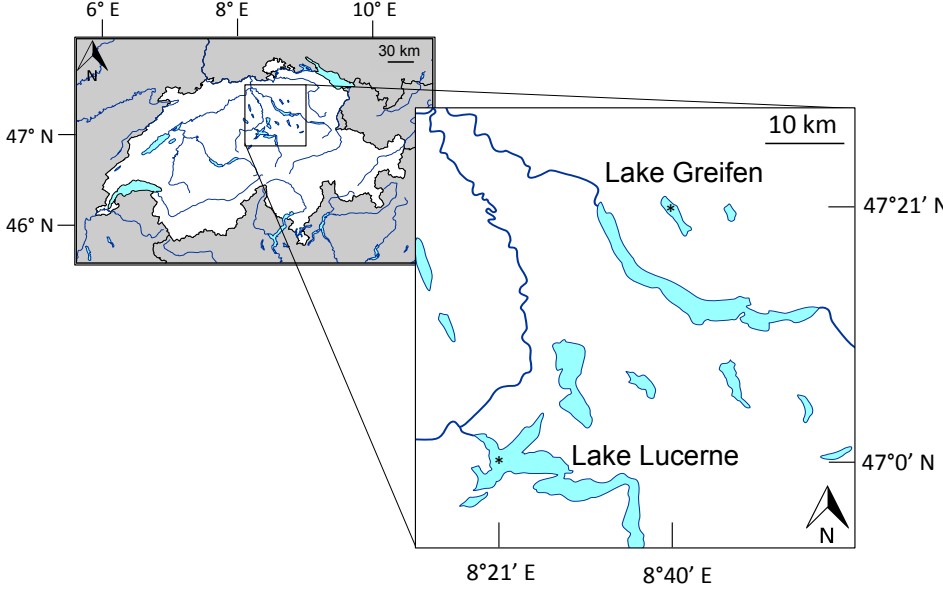

**Figure 1:** Map of Switzerland with locations of Lake Greifen and Lake Lucerne indicated. The locations where samples were collected are marked with the symbol *. Base map from d-maps (http://www.d-maps.com/carte.php?num_car=2648&lang=en).



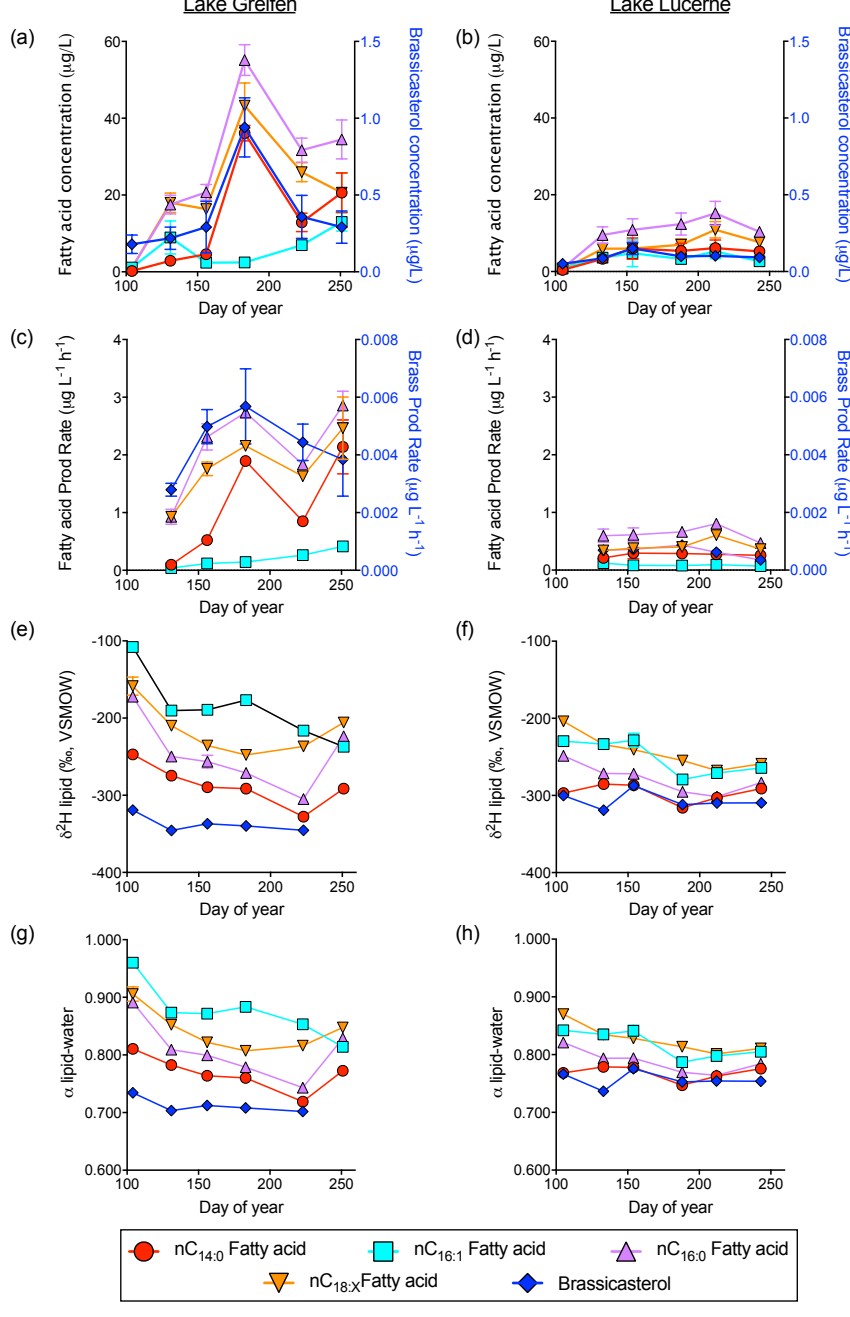

**Figure 2:** Time series of lipid concentrations in µg/L (panels a and b), lipid production rates in µg L$^{-1}$ h$^{-1}$ (panels c and d), lipid δ$^2$H values in ‰ relative to VSMOW (panels e and f), and α$_{lipid-water}$ values (panels g and h) for Lake Greifen (left column) and Lake Lucerne (right column) during the spring and summer of 2015. For panels a-d, brassicasterol concentrations and production rates are plotted on a separate y-axis in blue on the right of each panel. Error bars represent 1 standard deviation of replicate measurements, and are propagated to include uncertainties from multiple sources in calculated production rates and α$_{lipid-water}$ values. In cases where error bars are not visible, they are smaller than the marker size.





**Figure 3:** Relationships between $\alpha_{lipid\text{-}water}$ values and lake surface temperatures in Lakes Greifen and Lucerne throughout the spring and summer of 2015. Panel a is aggregate data from all fatty acids in each lake, b is $nC_{14:0}$ fatty acid, c is $nC_{16:0}$ fatty acid, d is $nC_{16:1}$ fatty acid e is unsaturated $C_{18}$ fatty acids, and f is brassicasterol. In each panel, pink squares represent data from Lake Greifen and blue circles represent data from Lake Lucerne. Error bars are propagated 1σ uncertainty from replicate measurements of surface water and lipid $\delta^2H$ values. In cases where error bars are not visible, they are smaller than the marker size. Dashed lines represent 95% confidence intervals of the linear regression. Statistics associated with each curve are summarized in Table 2.



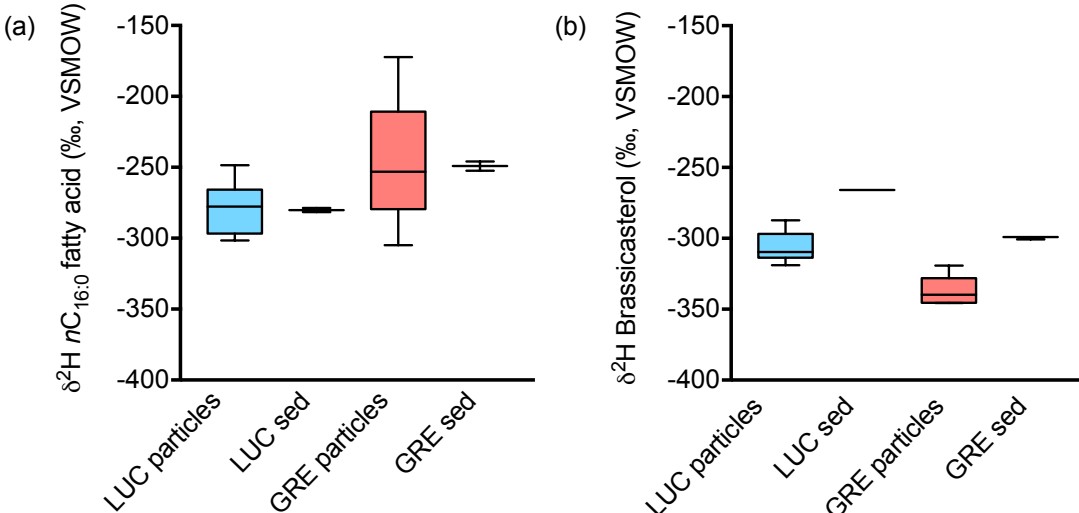

**Figure 4:** Comparison of $\delta^2H$ values of lipids collected by filtering suspended particles from surface water with lipids in surface sediments and sediment traps in Lake Lucerne (blue) and Lake Greifen (pink). Panel a shows values for $nC_{16:0}$ fatty acid and panel b shows values for brassicasterol.



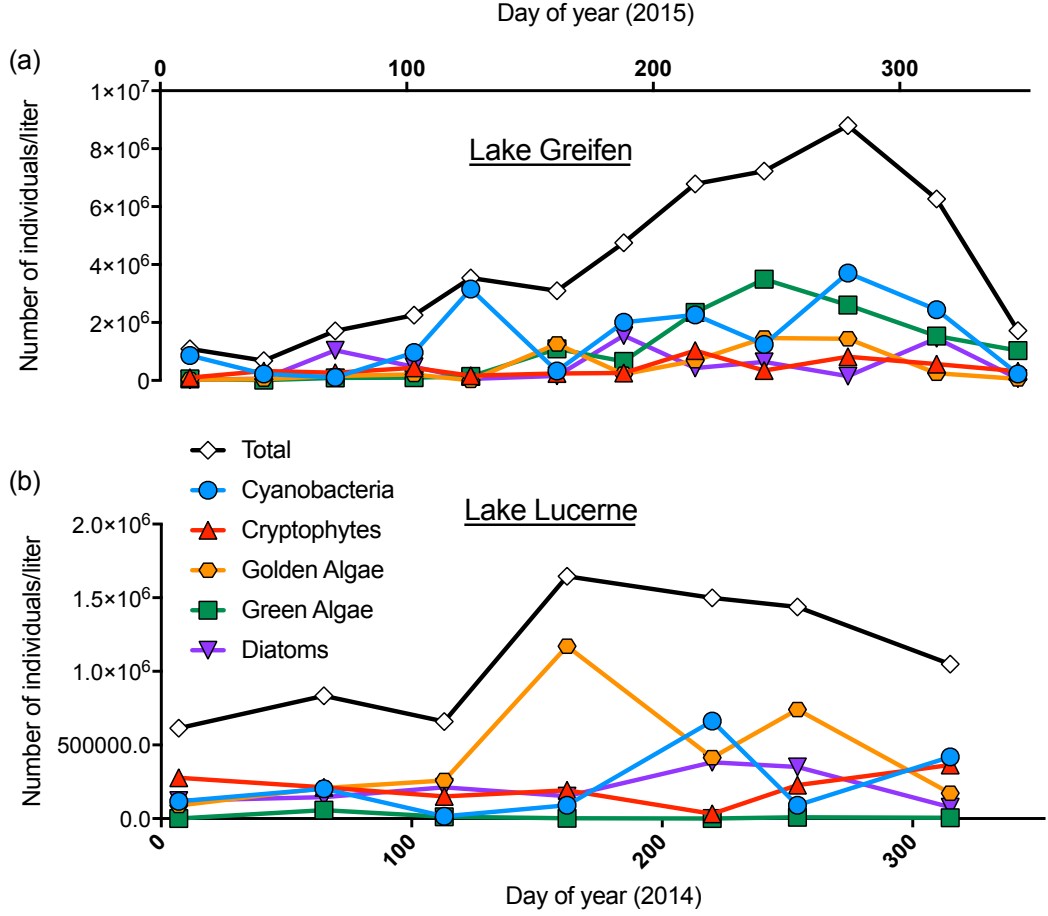

**Figure 5:** Cell counts (individuals per liter) for all algae and for most common taxa of algae in (a) Lake Greifen throughout 2015 and (b) Lake Lucerne in 2014. The scale of the y-axis differs between the two panels. Data from long-term monitoring program run by the department of Aquatic Ecology at Eawag.

