# Peer review of "Interplay of temperature, productivity, and community assemblage on hydrogen isotope signatures of algal lipid biomarkers"

_Biogeosciences, 2017_

## Referee Comment (RC1) · Anonymous Referee #1 · 14 Mar 2017

Summary: Ladd et al. describes the lipid composition and hydrogen isotopes of particulate OM through a time series from alpine two lakes and relates observed variability to temperature and nutrient availability. The work is strengthened by concurrent labeled incubations to get at lipid production rates and comparison to recent sediments. This work is robust, the conclusions are generally well supported, and the paper is well written.

I would like to see more data from the sedimentary samples and a revision in terms used to describe fractionation, but otherwise recommend that this manuscript be accepted for publication.

Larger comments: Probably my largest criticism of this manuscript is in somewhat

confusing usage of terms that descript the magnitude and direction of isotope fractionation. The most common usage in this manuscript is 'increase/decrease in hydrogen isotope fractionation' which is inherently vague because it doesn't indicate direction. Describing alpha values as high and low is equally problematic as a very low alpha describes a very large fractionation ( albeit a very large negative fractionation or depleted lipid signature). I would suggest converging on a single nomenclature and describing pools as more or less D-enriched or D-depleted and fractionations as positive, negative, more negative etc, rather than using term like higher and lower and increased and decreased. Section 4.1.1 is a good example of where you switch fluidly between XX permil per degree C to alpha in 0.001 and back with concurrent usage of increase, decrease, etc.

Please differentiate the cyanobacteria from eukaryotic algae. Lumping the two is a vestige from the time prior to DNA sequencing technologies that allowed for easy differentiation. This is imprecise and unnecessary for this manuscript. Please revise.

p10 Section 3.3 – Why do we only get to see data for the c16:0 FA for the sedimentary lipid section? Do the other lipids agree? I would also like to see if the sediment traps and the core top data agree with one another. Given the accumulation rate and sampling procedure the differences might suggest seasonal signals vs. annual integration. Generally the level of data and detail for the sedimentary samples should be as robust as the particulate (given that you are comparing the two and the later is the signal that most people look at).

p12-15 – As you have both cell counts and lipid yield data you should be able to normalize for cell counts in this discussion. You note at the end of the paragraph that cell counts agree, but it seems like you can take it a step farther.

p15-7 – I don't buy this explanation and you definitely haven't demonstrated that this is a general feature of isoprenoids in your samples. Presumably there is plenty of phytol in your extracts, what does that look like?

p16-1 – Can you calculate and compare a weighted average isotope value (as in Osburn et al., 2016) between these two?

I am generally curious about what other compounds were present in this dataset that can be compared between the particulate and the sediments. In particularly the midchain fatty acids as these are often used to describe paleolake water.

Figure 3: I am not sure all of these subfigures are useful. A bigger stacked figure showing all compounds from a single lake might be more informative.

Figure 4: Divide out sediment traps and surface sediment data then report all measured rather than just C16:0

Smaller comments p4-26-32 – How permeable were the incubating carboys to light and to air? It seems like the experiment is meant to replicate growing conditions, but doesn't really account for change associated with different light conditions or lack of oxygen. We can't really evaluate how comparable the incubations were to normal conditions. p5-33 – This is cooler and a shorter time than I have seen before. Is there reference for this? If not, are you sure that this condition was sufficient to quantitatively extract all lipids? p6-5-8 – You don't report compositional information for other lipids. This is ok, but is there anything noteworthy? p6-15 – What compound classes are eluting in each of these fractions? p6-22 – Do any of the aforementioned methods need citations or are they all original to this study? p7-24 – Please confirm that reported errors include propagated errors from replicate measurements, standards, and derivatization processes. p8 section 3.1 – this section could probably be condensed, it is quite repetitive. p9-2 – lowest here means fastest right? I would suggest going with time designations when referring to residence times p11, sec 4.1.2. this has been studied in plants p12-29 – lower alpha is confusing here p13-2 – The ultimate explanation should be the same for all FAs unless you are suggesting a different source. p13-11 – This is a bit of an overstatement. Just because we don't understand the governing mechanisms of species specific fraction does not make it random. p13-27 – Can

a statistical analysis help you here? Does any taxa change correlate with changes in fractionation? p13-31&32 – This is a false comparison, there are photoheterotrophs in this paper and they look far more like photoautotrophs than they do heterotrophs. You must at least acknowledge the photoheterotrophs. p14-2 – This hypothesis is also not consistent with an oligotrophic lake either. many algae and cyanobacteria are capable of mixotrophy. Almost all of them do this at night and some under diverse conditions. p14-18 – Be careful, there are plenty of bacteria that make primarily 16:0, 16:1, and 18:1. p17-14 – Reporting an offset in permil would be helpful to the paleo community p17-19 – isoprenoid change to sterol, you haven't made the case for isoprenoids more generally.

Minor editorial comments: p1-9 – isotope → isotopic p1-21 – in situ should be italicized here and throughout the manuscript p1-22 – increased magnitude of fractionation needs a direction term as the fractionation factor for lipid-water can be both above and below 1. p2-22-23 – This is stating the obvious, rephrase or combine with the following sentence. p2-25 – also metabolism see Zhang et al., 2009 or Osburn et al. 2016 p3-1 – I think that should be permil per degree C p3-5 – I am not sure what 'more' fractionation means p3-13 – remove 'the' p4-10 – What is the depth at this sampling point? p5-24&25 – something is wrong with these sentences. Perhaps a 'was' after 'sample' p7-10 – Initially? Were they subsequently converted to a different scale? p7-31 13 needs superscript p7-32 – lipid p8-15-16 – these numbers are the same (1.0 and 1.1) but the text suggests that one should be lower. p9-19 – add 'relative' after depleted p14-26 space before 2010
* * *

---

## Referee Comment (RC2) · R.H. Smittenberg (Referee) · 16 Mar 2017

Review Ladd, Dubois & Schubert: Interplay of temperature, productivity [...] lipid biomarkers

The authors measured the hydrogen isotopic composition (d2H) of lipid biomarkers, in particular short-chain fatty acids (C14-C18) and brassicasterol, from particulate organic matter filtered on a monthly basis from the surface water of two Swiss lakes, over the course of the algal 'growing season' of 2015. They combined these measurements with estimates of productivity using 13C labeling and data of community assemblage, and other environmental data like temperature and trophic status of the two lakes - one eutrophic and the other oligotrophic. This study gives useful insights in the hydrogen isotopic fractionation during biosynthesis of these lipids through time, the factors that influence this fractionation. The study is a welcome expansion of similar efforts performed on algal cultures, and aids in the assessment under what conditions biomarker d2H can potentially be used as a sedimentary proxy for past hydrological changes, and/or what the limitations are. The study is set up and executed properly, and the paper is well written. I have, however, some remarks, comments and questions I like the authors to address.

* Page4 Line26. I need to assume the carboys were made of clear plastic to allow photosynthesis?
* P8L15 increased from April to July? Or levels were low from April to July? Write more clearly.
* P9L24-25 "When analyzing.. Table 2)". Unclear sentence, rewrite.
* P10L31-34 "The slope....  " Not clear, rewrite.
* P11L30-33. This part appears out of place and fits better within the next section
* P12. Section 4.1.3. Lipid production rate.  -  I suggest to rename this section to 'trophic conditions' or 'nutrient availability', which is a primary environmental factor similar to temperature and light - with all three bearing on productivity and related fractionation. At the moment the discussion appears a bit mixed, nutrient availability and growth rate are somewhat used interchangeably.
* About the source of the fatty acids: The authors appear to only consider algae, or at any case aquatic organisms, as their source. However, fatty acids may also come from terrigenous sources, and this potential source may change over time. For example, surface runoff during early spring may bring in relatively large amounts of terrestrial organic matter at a time that lake primary productivity is still low.
* The non-existent correlation between fractionation factor and growth rate is likely due to the surprising low growth observed at day 220 at lake Greifen (why so much lower than at day 180, do the authors have an explanation?), and with just 5 measurements over the entire period this is bound to give bad statistics. I therefore wonder if there is no other information available about algal productivity, possibly the data from the long-term monitoring program at EAWAG could be used? Have the authors considered a more simple method of estimating productivity like chlorophyll concentration? How dependable and reproducible is the labeling-incubation method? What if the productivity data point at day 220 (and even the concentration of FA) from lake Greifen is compromised - would a higher rate at day 220 suddenly result in a good correlation?
* To what extent do turnover time and export of dead organic matter (or lack thereof) may have an influence on the bulk hydrogen isotopic compositions measured? How much algal biomass is taken up by heterotrophs and recycled, thereby partially

keeping the original isotopic signature? How much particulate OM is alive? In other words, how much 'memory' does the system have over the season leading to attenuation of the isotopic signal? If there is such attenuation, then the instantaneous productivity at a given point in time, especially at a later stage when it is going down, may be ever more unrelated to the accumulated particulate OM and lipid stock. Note that lake temperature has a large inertia thus will automatically correlate well with any other parameter with a slow response time.

* On page 14, the authors argue against a large contribution of heterotrophic bacteria based on low abundances of iso- and anteiso fatty acids. However, a large amount of heterotrophic biomass might be planktonic and not bacterial, while also not all heterotrophic bacteria will produce exactly those biomarkers - the majority will still predominantly produce C16:0 FA. It is not clear from the text to what extent the presented algal community data reflects only phototrophic algae (it is presented as such), or if these data are more inclusive to all microbial life (in which case heterotrophic plankton is surprisingly absent).

* P13L32. I would be very careful assuming that all heterotrophs have more enriched fatty acids than phototrophs based on only one study.

* P15L11-15. It is very well possible, or even likely, that the different lakes (with quite different trophic status) host different diatom species (or even non-diatoms, who knows..) making brassicasterol. Zhang et al has shown that different species making the same lipid may fractionate quite differently. This may also explain the large difference in fractionation of brassicasterol in the two different lakes.

* It would be useful to plot temperature through the season - not one based on five own measurements, but those from EAWAG or a similar service.

---

## Author Comment (AC1) · 9 Apr 2017

*Thank you for your comments and the helpful feedback. We have responded to each of your points in line with your text below. Our responses begin with the open bullet points. For the most part we agree with your suggestions, and have indicated our plans for a revised manuscript, which we will complete after the discussion period ends. In a few cases, we think that further dialogue would be helpful, either because we are unclear about the intent of your comment, or because we have a different perspective. We have also added some points of clarification to questions you have.*

==We have highlighted text below in blue where we think additional feedback from you would be helpful as we complete our revisions to the manuscript.==

*Thanks again for your time and feedback,*

*Nemiah Ladd*

Summary: Ladd et al. describes the lipid composition and hydrogen isotopes of particulate OM through a time series from alpine two lakes and relates observed variability to temperature and nutrient availability. The work is strengthened by concurrent labeled incubations to get at lipid production rates and comparison to recent sediments. This work is robust, the conclusions are generally well supported, and the paper is well written.

I would like to see more data from the sedimentary samples and a revision in terms used to describe fractionation, but otherwise recommend that this manuscript be accepted for publication.

Larger comments:

- Probably my largest criticism of this manuscript is in somewhat confusing usage of terms that descript the magnitude and direction of isotope fractionation. The most common usage in this manuscript is 'increase/decrease in hydrogen isotope fractionation' which is inherently vague because it doesn't indicate direction. Describing alpha values as high and low is equally problematic as a very low alpha describes a very large fractionation (albeit a very large negative fractionation or depleted lipid signature). I would suggest converging on a single nomenclature and describing pools as more or less D-enriched or D-depleted and fractionations as positive, negative, more negative etc, rather than using term like higher and lower and increased and decreased. Section 4.1.1 is a good example of where you switch fluidly between XX permil per degree C to alpha in 0.001 and back with concurrent usage of increase, decrease, etc.
    - Thanks for the feedback. It can often be difficult to tell what will be confusing for other readers when looking at your own writing. You are correct that this could be improved. Based on your comment here and on some of the detailed comments below, it seems like some of the biggest problems stem from the fact that more fractionation means smaller $\alpha$ values when $\alpha < 1$, as is typically the case for net fractionation during lipid synthesis by photoautotrophs. We did not adequately account for the possibility of $\alpha > 1$ in our usage of this term, and that also caused some points to be unclear. We will revise the

paper carefully with this advice in mind.

- o Another source of confusion is that it is often more intuitive to talk about fractionation in per mil terms (since this is what we report our δ values in). We agree that it is important to make our usage more consistent, and will strive to do so in a revised draft.

- Please differentiate the cyanobacteria from eukaryotic algae. Lumping the two is a vestige from the time prior to DNA sequencing technologies that allowed for easy differentiation. This is imprecise and unnecessary for this manuscript. Please revise.
  - o Again, we agree that this is a valid point, and that the original manuscript is not precise enough. We will differentiate cyanobacteria and eukaryotic algae throughout the revised manuscript.

- p10 Section 3.3 – Why do we only get to see data for the c16:0 FA for the sedimentary lipid section? Do the other lipids agree? I would also like to see if the sediment traps and the core top data agree with one another. Given the accumulation rate and sampling procedure the differences might suggest seasonal signals vs. annual integration. Generally the level of data and detail for the sedimentary samples should be as robust as the particulate (given that you are comparing the two and the later is the signal that most people look at).
  - o When the sediment samples were originally analyzed, the concentrations were optimized for $nC_{16:0}$. In many cases, the peak areas were not appropriate for other fatty acids, and the standard deviations of those measurements were high. We are running these samples on the GC-IRMS again at a more suitable concentration in order to be able to include sediment data from all of the compounds discussed from the filters, and will include those data and an expanded discussion about the sediment data in our revised manuscript.
  - o We will also separate the core top and sediment trap data from each other in our revised version of Figure 4.

- p12-15 – As you have both cell counts and lipid yield data you should be able to normalize for cell counts in this discussion. You note at the end of the paragraph that cell counts agree, but it seems like you can take it a step farther.
  - o A problem with doing this is that the cell count samples were collected on different days than the lipid samples. We can linearly extrapolate between the monthly cell counts to come up with an estimate of the number of cells on our sampling dates and normalize the lipid yields to these. Do you think such an analysis would be helpful? Given that the cell counts were collected as part of a separate project that we did coordinate with in advance, is it possible to be more quantitative here, or is a qualitative comparison most appropriate?

- p15-7 – I don't buy this explanation and you definitely haven't demonstrated that this is a general feature of isoprenoids in your samples. Presumably there is plenty of phytol in your extracts, what does that look like?
  - o We have phytol in our extracts. It eluted in the first fraction of the silver nitrate columns we performed on our alcohol fraction (third fraction collected during our silica gel columns). We have not

measured phytol $\delta^2H$ values from these samples yet, but will try to do so in time to include in the revised manuscript.

- o We agree that we probably overstated our case by referring to all isoprenoids on the basis of brassicasterol measurements alone. We will scale back our claim and revise the text to remove the reference to all isoprenoids.
- o It is likely that phytol will exhibit a different response than sterols, given that it is almost always depleted in $^2H$ relative to sterols in the same organism. Possible reasons for this depletion are (1) the fact that phytol synthesis occurs in the chloroplast, rather than in the cytosol, as is the case for sterols or (2) phytol can be produced by the deoxy-D-xylulose phosphate (DOXP) pathway, rather than the mevalonic acid (MVA) pathway, which is most commonly used to synthesize sterols in the cytosol.

- p16-1 – Can you calculate and compare a weighted average isotope value (as in Osburn et al., 2016) between these two?
    - o We are generally not in favor of calculating weighted average isotope values. The differences in $\delta^2H$ values among different lipids are not trivial and can be related to real information about environmental conditions, lipid sources, and physiological state. This information is lost when computing weighted averages, so our preference is to report all compound specific isotope values individually.

- I am generally curious about what other compounds were present in this dataset that can be compared between the particulate and the sediments. In particularly the midchain fatty acids as these are often used to describe paleolake water.
    - o In addition to the compounds measured in this study, the filters contained very small amounts of relatively short (~C17) *n*-alkanes, but concentrations were not high enough for H isotope measurements. There was abundant phytol, some cholesterol and stigmasterol, and trace amounts of other sterols at concentrations too low for hydrogen isotopes. The stigmasterol split between silver nitrate fractions and was not measured for H isotopes. The cholesterol was not measured since it was assumed to be mostly produced by heterotrophs. As mentioned in response to a previous comment, it should be straight-forward to measure $\delta^2H$ values of the phytol, and we hope to include these data in the revised manuscript.
    - o There were only trace amounts of fatty acids longer than $C_{18}$, and none of the longer chain fatty acids or *n*-alkanes associated with leaf waxes were collected on the filters.

- Figure 3: I am not sure all of these subfigures are useful. A bigger stacked figure showing all compounds from a single lake might be more informative.
    - o Here is a modified version of Figure 3 following your suggestions. 95% confidence intervals are only included for the regressions that were significant at the $p < 0.05$ level. Including the statistics in on the stacked figure gets pretty messy, but they are all listed in Table 2. If

you have additional thoughts about the best way to show this data (it is a little busy with all the curves on top of each other), we would appreciate it. We would be inclined to replace Figure 3 with this one in the revised manuscript, and to move the old version of Figure 3 to the supplemental material, in case anyone wants to look at the data in more detail.

[Figure]

- Figure 4: Divide out sediment traps and surface sediment data then report all measured rather than just C16:0
    - We plan to do this for the revised paper, along the lines of our response to your third "larger comment."

Smaller comments
- p4-26-32 – How permeable were the incubating carboys to light and to air? It seems like the experiment is meant to replicate growing conditions, but doesn't really account for change associated with different light conditions or lack of oxygen. We can't really evaluate how comparable the incubations were to normal conditions.
    - The carboys were made from transparent plastic, and we will add this qualifier to the text.

- p5-33 – This is cooler and a shorter time than I have seen before. Is there reference for this? If not, are you sure that this condition was sufficient to quantitatively extract all lipids?
    - When we first bought our microwave extraction system, our lab manager performed a number of tests of different extraction programs using replicates of a large homogenized sediment sample from Lake

Zug in central Switzerland. These included a range of temperatures up to 100°C, and extraction times up to 40 minutes. All samples were extracted twice, and the replicate extractions were analyzed as well in order to assess whether the first extraction had successfully recovered the lipids from the sample. In this test, extracting at higher temperatures than 70°C and for longer times than 5 minutes did not have a significant effect on lipid yields.

- p6-5-8 – You don't report compositional information for other lipids. This is ok, but is there anything noteworthy?
  - Please see above response to 7th bullet point in the larger comments.

- p6-15 – What compound classes are eluting in each of these fractions?
  - We will update this sentence to read: "The first fraction, containing *n*-alkanols and phytol, was eluted with 20 mL of 4:1 hexane/DCM, the second fraction, containing stanols and singly unsaturated sterols (such as cholesterol) with 20 mL of 1:1 hexane/DCM, the third fraction, containing most doubly unsaturated sterols including brassicasterol, with 16 mL of DCM, and the remaining compounds with 4 mL of ethyl acetate."

- p6-22 – Do any of the aforementioned methods need citations or are they all original to this study?
  - The methods reported here reflect minor modifications of long-established protocols. For the most part, they are not unique to this study. For example, the saponification and silica gel protocols are identical to those used in several recent papers about lipids from the Biogeochemistry group at Eawag. However, those methods represent small tweaks on long-standing methods (volume of solvent used, brand of silica gel column, etc.) and do not represent a novel innovation. We observe that in many organic geochemical papers, authors cite their previously published work in the portion of their methods about lipid extraction and purification, even though those papers are not the original example of silica gel chromatography or base hydrolysis. We think it is more useful to provide the specifics of our protocols so that the reader can clearly see what we did, without having to search for additional papers.
  - The specific silver nitrate scheme used to purify brassicasterol in this study is new, but the concept of using silver nitrate impregnated silica gel chromatography to separate organic compounds based on their number of double bonds is long-standing and well-established.

- p7-24 – Please confirm that reported errors include propagated errors from replicate measurements, standards, and derivatization processes.
  - We will add the following text after "mass balance": "and reported errors represented propagated errors from replicate measurements and the uncertainties associated with the added hydrogen."

- p8 section 3.1 – this section could probably be condensed, it is quite repetitive.

- o Thanks for the feedback. We will condense this section in the revised manuscript.

- p9-2 – lowest here means fastest right? I would suggest going with time designations when referring to residence times
  - o We will change "lowest" to "shortest" and agree that it makes the point clearer

- p11, sec 4.1.2. this has been studied in plants
  - o The light effect on hydrogen isotope fractionation in plants has been attributed to changes in transpiration (e.g. Yang et al., 2009, doi:10.1007/s00442-009-1321-1), which is not applicable for algae and cyanobacteria.
  - o We are familiar with an experiment by Marc André Cormier and Ansgar Kahmen where biosynthetic fractionation in leaf waxes was measured for plants grown under different light conditions. This work has been presented at conferences and workshops, but is not yet published. Is this what you are referring to? If there is other work about changes in biosynthetic fractionation (not changes in net fractionation due to changes in transpiration) in plants grown at different light levels, we are not familiar with it, and would appreciate it if you could direct us to the appropriate papers.

- p12-29 – lower alpha is confusing here
  - o We will modify along the lines of your earlier comment.

- p13-2 – The ultimate explanation should be the same for all FAs unless you are suggesting a different source.
  - o Given that the nC16:1 FA concentrations do not covary with the others, and that there are significant changes in the ratio of nC16:0 to nC16:1, it seems possible that there is a different source for them, or that only a limited number of the nC16:0 producers also make nC16:1, and that their relative abundance varied throughout the study period. We can add this point in the revised manuscript.

- p13-11 – This is a bit of an overstatement. Just because we don't understand the governing mechanisms of species specific fraction does not make it random.
  - o We will change this line to "…varies by hundreds of per mil among species in ways that are not yet understood."

- p13-27 – Can a statistical analysis help you here? Does any taxa change correlate with changes in fractionation?
  - o Unfortunately, the two sample sets were not collected on the same day, since they come from two uncoordinated sampling campaigns. Is it useful to try to apply a statistical analysis to a linear interpolation the changing taxa? We can do a multiple linear regression to see if changes in taxa correlate with changes in fractionation, with that limitation in mind.

- o We have also arranged for pigment concentrations to be analyzed on replicate filters that we collected the same day as our lipid samples. While changes in pigment abundance are not as species-specific as the cell counts, they can perhaps help us tell if the cell-count samples missed a notable bloom of specific taxonomic group that occurred closer to our sampling date.

- p13-31&32 – This is a false comparison, there are photoheterotrophs in this paper and they look far more like photoautotrophs than they do heterotrophs. You must at least acknowledge the photoheterotrophs.
  - o We will revise this section to make the distinction between photoheterotrophs and heterotrophs more clear, and will also add in references to other work about variability in hydrogen isotope fractionation that is due to metabolic pathways (in particular, papers by Osburn et al. and Heinzelmann et al.).

- p14-2 – This hypothesis is also not consistent with an oligotrophic lake either. many algae and cyanobacteria are capable of mixotrophy. Almost all of them do this at night and some under diverse conditions.
  - o Our point was that if mixotrophy (more common in the oligotrophic lake) has a significant effect on $^2$H-fractionation, one would be expect it to result in relatively more enriched fatty acids. Since fatty acid fractionation factors are similar between the two lakes for most of the summer, and the fatty acids tend to be more enriched in the spring in the eutrophic lake but not the oligotrophic one, we would suggest that seasonal changes in mixotrophy are unlikely to account for the difference in the magnitude of the seasonal change in $^2$H fractionation between the two lakes. We will try to make this point more clearly in the revised manuscript.

- p14-18 – Be careful, there are plenty of bacteria that make primarily 16:0, 16:1, and 18:1.
  - o We acknowledge that heterotrophic bacteria can produce a large amount of 16:0, 16:1, and 18:1. We will modify the text of this section as follows: "If the source of so much $n$C$_{16:0}$ fatty acids was bacterial, it might be expected to correspond to increased concentrations of lipids associated with heterotrophic bacteria, such as iso- and anteiso-C$_{15:0}$ and C$_{17:0}$ (Perry et al., 1979; Volkmann et al., 1980). Since there are not significant amounts of these short-chain odd-carbon branched fatty acids in the particulate organic matter throughout the time series, including in the early spring, it seems less likely that such a large component of the even-carbon fatty acids is be derived from bacterial sources. However, this does not rule out greater contributions from mixotrophic algae relying on heterotrophy in the spring, nor from heterotrophic bacteria species that primarily produce short-chain, even-carbon fatty acids."

- p17-14 – Reporting an offset in permil would be helpful to the paleo community
  - o We will add this to the revised draft.

- p17-19 – isoprenoid change to sterol, you haven't made the case for isoprenoids more generally.
  - o We will change "isoprenoid" to "sterol"

Minor editorial comments:

- p1-9 – isotope → isotopic
  - o Changed

- p1-21 – in situ should be italicized here and throughout the manuscript
  - o The style guidelines for *Biogeosciences* indicate that common Latin phrases, including "in situ", should not be italicized, so we will leave this unitalicized.

- p1-22 – increased magnitude of fractionation needs a direction term as the fractionation factor for lipid-water can be both above and below 1.
  - o We will change this in line with our response to your first comment.

- p2-22-23 – This is stating the obvious, rephrase or combine with the following sentence.
  - o We will combine these two sentences into the following revised sentence: The offset between the hydrogen isotopic composition of lipids and source water is described by the apparent fractionation factor, $\alpha_{\text{Lipid-Water}} = ({}^2H/{}^1H_{\text{Lipid}})/({}^2H/{}^1H_{\text{Water}})$, or in ‰ terms by $\varepsilon_{\text{Lipid-Water}} = ((\delta^2H_{\text{Lipid}} + 1000)/(\delta^2H_{\text{Water}} + 1000) - 1)*1000$, where $\delta^2H = (({}^2H/{}^1H_{\text{Sample}})/({}^2H/{}^1H_{\text{VSMOW}}) - 1)*1000$.

- p2-25 – also metabolism see Zhang et al., 2009 or Osburn et al. 2016
  - o This is a good point and it is relevant for the subsequent discussion. We will add it to the text and to Table 1, and include two papers from Heinzelmann, 2015 in reference to this point as well.

- p3-1– I think that should be permil per degree
  - o Correct, we will modify to "‰ per °C"

- p3-5 – I am not sure what 'more' fractionation means
  - o We mean that alpha is further from unity; in this case, since the fractionation is more negative, alpha is smaller and that represents more fractionation. We will revise this sentence to make it clearer, in line with our response to your first comment.

- p3-13 – remove 'the'
  - o We will remove it

- p4-10 – What is the depth at this sampling point?
  - o The water depth was 96m. This information will be added to this section.

- p5-24&25 – something is wrong with these sentences. Perhaps a 'was' after 'sample'
  - We will add "was" after "sample"

- p7-10 – Initially? Were they subsequently converted to a different scale?
  - We will delete the word "initially," which is not relevant

- p7-31 13 needs superscript
  - We will change 13 to $^{13}$

- p7-32 – lipid
  - We will delete the "s" at the end of "lipids"

- p8-15-16 – these numbers are the same (1.0 and 1.1) but the text suggests that one should be lower.
  - The concentrations were the same as each other for the first sampling date in April (both ~1 μg/L), but subsequently, nC16:1 concentrations were much lower than those of nC16:0 (13 μg/L = maximum value for nC16:1, 55 μg/L for nC16:0)

- p9-19 – add 'relative' after depleted
  - We will add this

- p14-26 space before 2010
  - We will add this

---

## Author Comment (AC2) · 9 Apr 2017

*Dear Dr. Smittenberg,*

*Thank you for your comments and the helpful feedback. We have responded to each of your points in line with your text below. Our responses begin with the open bullet points. For the most part we agree with your suggestions, and have indicated our plans for a revised manuscript, which we will complete after the discussion period ends. In a few cases, we think that further dialogue would be helpful, either because we are unclear about the intent of your comment, or because we have a different perspective. We have also added some points of clarification to questions you have.*

*We have highlighted text below in blue where we think additional feedback from you would be helpful as we complete our revisions to the manuscript.*

*Thanks again for your time and feedback,*

*Nemiah Ladd*

The authors measured the hydrogen isotopic composition (d2H) of lipid biomarkers, in particular short-chain fatty acids (C14-C18) and brassicasterol, from particulate organic matter filtered on a monthly basis from the surface water of two Swiss lakes, over the course of the algal 'growing season' of 2015. They combined these measurements with estimates of productivity using 13C labeling and data of community assemblage, and other environmental data like temperature and trophic status of the two lakes - one eutrophic and the other oligotrophic. This study gives useful insights in the hydrogen isotopic fractionation during biosynthesis of these lipids through time, the factors that influence this fractionation. The study is a welcome expansion of similar efforts performed on algal cultures, and aids in the assessment under what conditions biomarker d2H can potentially be used as a sedimentary proxy for past hydrological changes, and/or what the limitations are. The study is set up and executed properly, and the paper is well written. I have, however, some remarks, comments and questions I like the authors to address.

- Page4 Line26. I need to assume the carboys were made of clear plastic to allow photosynthesis?
  - We will add the word "transparent" here to clarify

- P8L15 increased from April to July? Or levels were low from April to July? Write more clearly.
  - We will change this sentence to read: "Lipid concentrations increased significantly in Lake Greifen from April to July, and then declined slightly from July to September."

- P9L24-25 "When analyzing.. Table 2)". Unclear sentence, rewrite.
  - We will change this sentence to read: "Regressions for individual fatty acids typically had higher $R^2$ values than the pooled correlation for all short-chain fatty acids."

- P10L31-34 "The slope.... " Not clear, rewrite.
  - This sentence actually just repeats the same information that is included in the above bullet point from the results section. We will delete it from the revised version of the manuscript.

- P11L30-33. This part appears out of place and fits better within the next section
  - Based on this comment and the following one, we think it would be a good idea to reorganize the discussion slightly. We will keep sections 4.1.1 about temperature and 4.1.2 about light availability. We suggest adding a short section about trophic status/nutrient availability (4.1.3), and then a section about productivity (4.1.4). This section will briefly explain how the three environmental factors (temp, light, nutrients) relate to productivity and lipid production rates, and then discuss the lack

of correlation between production rates and $^2$H fractionation factors. As part of this rearrangement, we will move the information on these lines to the new section 4.1.4 about productivity.

- P12. Section 4.1.3. Lipid production rate. - I suggest to rename this section to 'trophic conditions' or 'nutrient availability', which is a primary environmental factor similar to temperature and light - with all three bearing on productivity and related fractionation. At the moment the discussion appears a bit mixed, nutrient availability and growth rate are somewhat used interchangeably.
    o We agree that it would be more precise to rename this section, and to clearly split the discussion about trophic status and lipid production rates. For the revised manuscript, we will restructure the discussion as described in response to the previous comment.

- About the source of the fatty acids: The authors appear to only consider algae, or at any case aquatic organisms, as their source. However, fatty acids may also come from terrigenous sources, and this potential source may change over time. For example, surface runoff during early spring may bring in relatively large amounts of terrestrial organic matter at a time that lake primary productivity is still low.
    o It is true that vascular plants also produce large amounts of short chain fatty acids. However, our samples were collected from surface water in the middle of the lake, and there is not a good mechanism to transport large amounts of terrestrial material to such a location. Our incubations indicate that short-chain fatty acids are produced rapidly relative to the standing stock of lipids in the surface water, as indicated by the short residence times reported in Table 3. This result suggests that the vast majority of the short-chain fatty acids collected on our filters are produced in the lake water. Additionally, this result argues against large contributions of fatty acids from heterotrophs, since it would take more time for the $^{13}$C label to be consumed after fixation by photoautotrophs, and our incubations only lasted for six hours.
    o Another reason why we think it is unlikely that there were significant contributions of short-chain fatty acids from non-aquatic sources is the absence of long-chain $n$-alkanoic acids, long-chain $n$-alkanes, and other biomarkers for higher plants on our filters.

- The non-existent correlation between fractionation factor and growth rate is likely due to the surprising low growth observed at day 220 at lake Greifen (why so much lower than at day 180, do the authors have an explanation?), and with just 5 measurements over the entire period this is bound to give bad statistics. I therefore wonder if there is no other information available about algal productivity, possibly the data from the long-term monitoring program at EAWAG could be used? Have the authors considered a more simple method of estimating productivity like chlorophyll concentration? How dependable and reproducible is the labeling-incubation method? What if the productivity data point at day 220 (and even the concentration of FA) from lake Greifen is compromised - would a higher rate at day 220 suddenly result in a good correlation?
    o We also find the low production rates from day 223 in Lake Greifen to be a bit confusing and unexpected, but they may have to do with the weather from that day. We tried to restrict our sampling to fully sunny days in order to minimize confounding effects from light availability. Unfortunately, day 223 on Lake Greifen ended up being partially cloudy, and these incubations represent the only ones that were not carried out in full sunshine, which may account for the lower production rates.
    o It is true that one questionable value could skew small number statistics. We checked the correlations between $\alpha_{Lipid-Water}$ and lipid production rate for Lake Greifen without the sample from day 223. There were not significant correlations for any of the five lipids. The $R^2$ value for $nC_{16:1}$ was still the highest, but declined from 0.84 to 0.81. The correlations for the other lipids improved slightly, but the $R^2$ values remain relatively low.
    o We agree that it would be good to supplement our lipid production rates with chlorophyll concentrations. We have additional filters that have been stored frozen since collection since each sampling day, and are currently analyzing them for pigments. We hope to have these data to include in the revised manuscript. The

abundance of various pigments associated with specific taxa of algae may also be a helpful constraint for the algal community side of the story.

    o   Finally, with regards to the comment about the reliability of the $^{13}$C incubations, the method has been successfully employed several times in marine settings (e.g. Popp et al., 2006, doi:10.1029/2005PA001165, Prahl et al., 2004, doi:10.1016/j.dsr.2004.12.001, Wolhowe et al., 2014, DOI: 10.1016/j.pocean.2013.12.001). Precision between our replicate incubations from the same day averaged $11 \pm 7\%$ of the production rate for Lake Greifen and $15 \pm 9\%$ of the production rate for Lake Lucerne. These uncertainties are shown in the error bars on the lipid production rates in Figure 2.

- To what extent do turnover time and export of dead organic matter (or lack thereof) may have an influence on the bulk hydrogen isotopic compositions measured? How much algal biomass is taken up by heterotrophs and recycled, thereby partially keeping the original isotopic signature? How much particulate OM is alive? In other words, how much 'memory' does the system have over the season leading to attenuation of the isotopic signal? If there is such attenuation, then the instantaneous productivity at a given point in time, especially at a later stage when it is going down, may be ever more unrelated to the accumulated particulate OM and lipid stock. Note that lake temperature has a large inertia thus will automatically correlate well with any other parameter with a slow response time.

    o   Part of the reason why we did the incubations to measure lipid production rates was to address this question. For the most part, lipid production rates are high relative to the lipid concentrations, indicating quick turnover. Residence times for each target lipid are reported in Table 3, and for most cases are less than one day. The fatty acid with the longest residence times is $nC_{16:1}$, but even these never exceed three days. Admittedly, the production rates are based on incubations conducted during daylight hours only, so they are likely to average to a lower rate over a full 24-hour period. However, they suggest that most of the fatty acids in the lake are produced within the past week at the most.

    o   The compound with the longest residence times is Brassicasterol (Table 3). This is also the compound whose hydrogen isotope fractionation shows the smallest correlation with temperature, suggesting that the correlation between fatty acid $^2$H fractionation and temperature is not an artifact caused by two parameters both with slow response times.

- On page 14, the authors argue against a large contribution of heterotrophic bacteria based on low abundances of iso- and anteiso fatty acids. However, a large amount of heterotrophic biomass might be planktonic and not bacterial, while also not all heterotrophic bacteria will produce exactly those biomarkers - the majority will still predominantly produce C16:0 FA. It is not clear from the text to what extent the presented algal community data reflects only phototrophic algae (it is presented as such), or if these data are more inclusive to all microbial life (in which case heterotrophic plankton is surprisingly absent).

    o   In response to a similar comment from Reviewer 1, we have decided to modify the text at the end of the last paragraph of section 4.1.4 (starting at P14L17 of the original manuscript) to the following text: "If the source of so much $nC_{16:0}$ fatty acids was bacterial, it might be expected to correspond to increased concentrations of lipids associated with heterotrophic bacteria, such as iso- and anteiso-$C_{15:0}$ and $C_{17:0}$ (Perry et al., 1979; Volkmann et al., 1980). Since there are not significant amounts of these short-chain odd-carbon branched fatty acids in the particulate organic matter throughout the time series, including in the early spring, it seems less likely that such a large component of the even-carbon fatty acids is be derived from bacterial sources. However, this does not rule out greater contributions from mixotrophic algae relying on heterotrophy in the spring, nor from heterotrophic bacteria species that primarily produce short-chain, even fatty acids."

- P13L32. I would be very careful assuming that all heterotrophs have more enriched fatty acids than phototrophs based on only one study.

    o   Subsequent studies (such as Osburn et al., 2011, doi.org/10.1016/j.gca.2011.05.038, Heinzelmann et al., 2015a, doi: 10.1093/femsle/fnv065, Heinzelmann et al., 2015b,

10.3389/fmicb.2015.00408) have also observed more enriched fatty acids in heterotrophs relative to autotrophs. However, it is true that to date hydrogen isotope fractionation has only been investigated in a tiny portion of the myriad heterotrophic microbes (and photoautotrophs) that exist, and it is possible that there is more diversity in heterotroph hydrogen isotope fractionation than these initially studies indicate. We will modify this sentence to reflect this uncertainty, and to include the additional references.

- P15L11-15. It is very well possible, or even likely, that the different lakes (with quite different trophic status) host different diatom species (or even non-diatoms, who knows..) making brassicasterol. Zhang et al has shown that different species making the same lipid may fractionate quite differently. This may also explain the large difference in fractionation of brassicasterol in the two different lakes.
  - o It is true that different species can have different fractionation factors for the same lipid. The species assemblage data collected by Eawag can help us assess how much the diatom community varies between the two lakes. No diatom species were identified in Lake Lucerne in 2014 that were not present in Lake Greifen in 2015. However, some of the prominent diatom taxa from Lake Greifen are not present in Lake Lucerne. The most abundant of these are *Stephanodiscus* sp. (up to 25% of the diatom community in Greifen), followed by *Melosira* (up to 5% of the diatom community in Greifen). We will modify this section of the discussion to indicate that the differences in brassicasterol net fractionation between the two lakes may be due to contributions from different diatom producers. This is almost certainly the case for fatty acids between the two lakes as well.

- It would be useful to plot temperature through the season - not one based on five own measurements, but those from EAWAG or a similar service.
  - o In addition to our six measurements from Lake Greifen, the environmental agency for Canton Zurich measured lake water temperature at monthly intervals. Their data is plotted in black in the figure below, and generally agrees well with our data (in red) from slightly different days. We are trying to find similar data from Lake Lucerne, and will add lake surface temperature curves to figure 2 in the revised version of the manuscript.

---

## Author Response (AR1)

Dear Dr. van der Meer,

Thank you for your helpful comments and feedback on our manuscript "Interplay of community dynamics, temperature, and productivity on the hydrogen isotope signatures of lipid biomarkers." We have revised the manuscript in accordance with your suggestions and the feedback from the two external reviewers.

There are two major changes to the revised manuscript. First, following your suggestion, we have expanded our focus to consider the influence of heterotrophic sources of fatty acids. We have reframed the introduction to focus on the use of generic fatty acid hydrogen isotopes as a proxy for net community metabolism, as well as the use of phytoplankton-specific biomarkers as a water isotope proxy. We have rearranged the discussion to focus first on the effect of lipid source on our observed  $\alpha_{Lipid-Water}$  values (new section 4.1) and then on the effect of environmental gradients on  $\alpha_{Lipid-Water}$  in phytoplankton (section 4.2). To accommodate the expanded discussion of heterotrophic fatty acids, we have condensed the discussion about light availability, which is unlikely to have had a major effect in lake surface waters during spring/summer.

The other major change is that we have removed the data and discussion pertaining to the sediment trap and core top samples. Following the advice of Reviewer 1, we initially tried to incorporate values from the remaining fatty acids, which we did not have good data for at the time of the original submission. There are interesting trends here, especially the 2H-enrichment of the core top fatty acids relative to those in sediment traps (see figure in updated response to reviewer 1). However, the additional discussion necessary to explain these trends expanded the paper's scope and reduced its focus. We believe it is more appropriate to include these data in a second manuscript we have been preparing concurrently, which also includes newly obtained sediment trap and core top data from 8 additional lakes in central Switzerland. Please see our updated response to Reviewer 1 for more of this data and the rationale for removing the sediment data from this submission.

We have made a number of smaller changes to the text, in line with the suggestions of the two reviewers. These are detailed in our updated responses to the two reviewers. We have also measured  $\delta^2$ H values of phytol, and updated the figures and discussion to include these results.

We believe that all of these changes have strengthened the manuscript considerably, and appreciate the time you and the reviewers have spent to improve this paper.

Best wishes,

Nemiah Ladd, on behalf of all co-authors

**Revised Response to Reviewer 1**

Thank you for your comments and the helpful feedback. We have responded to each of your points in line with your text below. Our responses begin with the open bullet points. For the most part we agree with your suggestions, and have indicated the changes we made in the revised manuscript. In a few cases, we disagreed with the suggestion, and have explained our reasoning below each point.

**Thanks again for your time and feedback,**

**Nemiah Ladd**

Summary: Ladd et al. describes the lipid composition and hydrogen isotopes of particulate OM through a time series from alpine two lakes and relates observed variability to temperature and nutrient availability. The work is strengthened by concurrent labeled incubations to get at lipid production rates and comparison to recent sediments. This work is robust, the conclusions are generally well supported, and the paper is well written.

I would like to see more data from the sedimentary samples and a revision in terms used to describe fractionation, but otherwise recommend that this manuscript be accepted for publication.

Larger comments:

- Probably my largest criticism of this manuscript is in somewhat confusing usage of terms that descript the magnitude and direction of isotope fractionation. The most common usage in this manuscript is 'increase/decrease in hydrogen isotope fractionation' which is inherently vague because it doesn't indicate direction. Describing alpha values as high and low is equally problematic as a very low alpha describes a very large fractionation (albeit a very large negative fractionation or depleted lipid signature). I would suggest converging on a single nomenclature and describing pools as more or less D-enriched or D-depleted and fractionations as positive, negative, more negative etc, rather than using term like higher and lower and increased and decreased. Section 4.1.1 is a good example of where you switch fluidly between XX permil per degree C to alpha in 0.001 and back with concurrent usage of increase, decrease, etc.
  - We have revised the text and now refer to changes in fractionation as increases or decreases in  $\alpha_{\text{Lipid-Water}}$ , and referenced these as decimals instead of %. We have also described changes in d2H values of specific pools as more 2H-depleted or 2H enriched.
- Please differentiate the cyanobacteria from eukaryotic algae. Lumping the two is a vestige from the time prior to DNA sequencing technologies that allowed for easy differentiation. This is imprecise and unnecessary for this manuscript. Please revise.
  - We have modified the text to refer to "cyanobacteria and eukaryotic algae" in place of the more generic "algae". In some cases, especially when comparing organisms with different core metabolisms, we found

it less clunky to lump these two groups together, but have used the terms "phytoplankton" or "photoautotrophs" instead of "algae."

- p10 Section 3.3 Why do we only get to see data for the c16:0 FA for the sedimentary lipid section? Do the other lipids agree? I would also like to see if the sediment traps and the core top data agree with one another. Given the accumulation rate and sampling procedure the differences might suggest seasonal signals vs. annual integration. Generally the level of data and detail for the sedimentary samples should be as robust as the particulate (given that you are comparing the two and the later is the signal that most people look at).
  - When the sediment samples were originally analyzed, the concentrations were optimized for  $nC_{16:0}$ . In many cases, the peak areas were not appropriate for other fatty acids, and the standard deviations of those measurements were high. We have rerun these samples on the GC-IRMS again at a more suitable concentration in order to be able to get good sediment data from all of the compounds discussed from the filters. The results show a more complicated picture than what was apparent when we had only measured palmitic acid, and pulling out the core top data from the sediment traps adds more complexity. Here is what the new data look like:

These have a number of interesting features, including 2H enrichment of most coretop fatty acids relative to sediment traps, and relatively good agreement between phytol  $\delta$ 2H values in POM and core tops. However, discussing possible reasons for these trends added quite a bit of text to the revised manuscript, and reduced its focus. Since submitting the first version of this paper to *Biogeosciences*, we have also measured  $\delta$ 2H values of multiple lipids from coretops and sediment traps in 8 additional lakes in central Switzerland. These samples were also collected in the

spring of 2015, but there is no corresponding time series of particulate organic matter from surface waters in these lakes. The full set of samples from ten lakes allows for a more robust discussion about differences in  $\delta$ 2H values between traps and core tops, and for differences in  $\delta$ 2H values among different types of lipids. Here is a summary figure of the fractionation between lipids in sediment traps and coretops:

The scatter plots below show the relationships between  $\delta 2H$  lipid and  $\delta 2H$  water for a subset of these lipids, as well as changes in  $\alpha_{\text{Lipid-Water}}$  with total phosphorus concentrations: